# Binding properties of sulfur to enable solvent-free fabrication of high-performance polymer-free sulfur-carbon positive electrodes

Yuhui An[1,7], Kyungbae Kim[1,7], Yun-Jeong Lee[2], Soyeon Ko[3], Faizan Ejaz[4], Yongming Liu[4], Beomjin Kwon[4], Seung-Ho Yu[2,5] ✉ & Yoon Hwa[6] ✉

The development of a scalable, cost-effective and environmentally benign manufacturing of binder-free sulfur positive electrodes can make substantial advance for lithium-sulfur (Li∥S) batteries as sustainable competitors to lithium-ion systems. Here we show a solvent- and binder-free method to fabricate sulfur-carbon composite electrodes directly on aluminum foil via thermal-assisted dry pressing. A key finding is the role of sulfur as a structural binder, where its softening, distribution, and adhesion properties enable the formation of mechanically robust electrodes without polymer binders. Systematic experimental characterizations and computational modeling reveal the underlying mechanisms governing electrodes formation and electrochemical performance. The developed binder-free positive electrodes achieve a reversible capacity of 932 mAh g$^{-1}$ after 500 cycles at 1.0 C rate; for comparison, conventional slurry-cast positive electrodes containing 10 wt.% binder deliver lower capacity under the same electrochemical test conditions. This scalable process has the potential to reduce fabrication costs by a factor of 2.1 while eliminating hazardous solvents and binders, offering a sustainable and cost-effective approach to advancing Li∥S battery technology.

The relentless pursuit of advanced energy storage systems stands at the forefront of modern scientific inquiry, driven by the urgent need for sustainable, efficient, and high-energy density solutions. Among the various contenders, lithium-sulfur (Li∥S) batteries have gained significant attention due to their high theoretical specific energy (2600 Wh kg$^{-1}$) and cost-effectiveness, primarily attributed to the abundance of sulfur, used as a positive electrode active material[1,2]. Despite the problematic issues associated with poor electronic conductivity of sulfur, soluble lithium polysulfide (Li-PS) intermediates such as Li-PS shuttling[3] and repeated changes in the porous structure of sulfur positive electrodes[4], diminishing cycling stability of Li∥S batteries, significant progress in enhancing the performance of Li∥S batteries accomplished by researchers have positioned Li∥S batteries to be one of promising future energy storage options[5–8].

Li∥S batteries typically employ the conventional slurry casting method to fabricate a sulfur positive electrodes with the incorporation of polymer binders (about 10 wt.% in general)[9]. The slurry casting process requires a processing solvent, typically N-Methyl-2-

[1]Materials Science and Engineering, Fulton School of Engineering, Arizona State University, Tempe, AZ, USA. [2]Department of Chemical and Biological Engineering, Korea University, Seoul, Republic of Korea. [3]Chemical Engineering, Fulton School of Engineering, Arizona State University, Tempe, AZ, USA. [4]Mechanical and Aerospace Engineering, Fulton School of Engineering, Arizona State University, Tempe, AZ, USA. [5]Department of Battery-Smart Factory, Korea University, Seoul, Republic of Korea. [6]School of Electrical, Computer and Energy Engineering, Arizona State University, Tempe, AZ, USA. [7]These authors contributed equally: Yuhui An, Kyungbae Kim. ✉e-mail: seunghoyu@korea.ac.kr; Yoon.Hwa@asu.edu

Pyrrolidone (NMP) to dissolve the binder and disperse the solid constituents uniformly. However, the use of solvent introduces environmental challenges and increases production costs due to the energy-intensive solvent evaporation and vapor recovery process, thereby undermining the sustainability and economic viability of Li‖S batteries[10,11]. Note that the electrode manufacturing process is the most energy-intensive part of battery manufacturing[12], with the wet-slurry coating, drying, and NMP solvent recovery making up ~50% of total electrode manufacturing costs[13,14]. The drying step accounts for about 39% of energy consumption, releasing ~1000 kg $CO_2$-equivalent emissions for every 10 kWh of batteries produced[15,16]. Eliminating the NMP recovery system could yield substantial benefits for battery manufacturing facilities, such as reducing capital costs by up to $6 million and saving 16.5% in areal footprint for a gigafactory producing GWh-level batteries[17]. As shown in Table S1, the estimated manufacturing cost of our dry process approach ($38.4 per kg of sulfur in the positive electrodes) is less than half the cost of a slurry-cast sulfur positive electrodes ($80.91 per kg of sulfur in the positive electrodes). Although aqueous binders such as styrene–butadiene rubber-carboxymethyl cellulose[18] and polyacrylic acid[19] can partially mitigate the environmental and economic impacts of electrode fabrication process, they introduce other challenges. For instance, aqueous binders often lead to electrode cracking during the slurry drying process[20], and leave residual water in positive electrodes. This residual water is challenging to remove through vacuum drying at the elevated temperature (typically 120–150 °C for conventional non-sulfur electrodes) due to low melting point of sulfur, which negatively impacts the performance of Li‖S batteries[20]. Recent reviews have highlighted growing interest in dry-fabrication of sulfur positive electrodes for Li‖S batteries as a means to improve manufacturing efficiency and interface control[21–24]. Moreover, although the role of binders is crucial in enhancing the mechanical durability of electrode films, binders increase the resistance in positive electrodes as they are generally electronic and ionic insulators[25]. Efforts to eliminate binders from sulfur positive electrodes have explored the use of three-dimensional porous carbon structure[26–29] and metal foam[30,31] as current collectors. However, prior binder-free strategies often rely on specialized or nonstandard current collectors that are incompatible with scalable manufacturing methods such as roll-to-roll coating, ultimately increasing production costs. In some cases, high-temperature melt-infiltration of sulfur into carbon substrates is also required[29], adding to process complexity and material expenses. Moreover, these approaches typically involve rigid or pre-structured porous carbon architectures where precise control over bulk porosity is difficult to achieve. These limitations make it challenging to implement such methods in high-throughput, cost-sensitive manufacturing environments, thereby restricting their scalability.

Building on this background, our research introduces a polymer binder- and solvent-free method for fabricating sulfur-carbon (S-C) composite positive electrodes using a conventional aluminum (Al) foil current collector. In this approach, sulfur serves a dual function as both the active material and an inherent binder, eliminating the need for conventional binders. By removing binders and processing solvents, our process provides an environmentally friendly, cost-effective alternative for electrode fabrication. The method is based on dry powder formation on Al foil followed by roll-pressing, both of which are compatible with continuous processing and allow control over key processing parameters such as areal loading and electrode porosity, highlighting its potential for scalable and tunable manufacturing. We conducted a comprehensive analysis of the structural, compositional, and electrochemical properties of the binder-free S-C composite positive electrodes using advanced techniques such as X-ray imaging, and in situ optical microscopy (OM). Our findings reveal the mechanisms driving the improved electrochemical performance, including structural features of binder-free sulfur positive electrodes that boost

electrochemical activity. This study demonstrates significantly enhanced electrochemical performance using inexpensive commercial porous carbon as a sulfur host in S-C composites, comparable to previous reports that relied on more expensive or complexly synthesized S-C composite active materials and conventional slurry-cast electrodes containing ~10 wt.% polymer binder. The binder-free S-C electrodes achieve a reversible capacity of 932 mAh g$^{-1}$ after 500 cycles, which is among the higher values reported in Table S2. These insights pave the way for more sustainable manufacturing practices for high-performance Li‖S batteries.

## Results and discussion

### Thermally-assisted consolidation of elemental sulfur

Substituting sulfur for a polymer binder in binder-free sulfur electrodes requires sulfur particles to bind together effectively. Given a low melting temperature of sulfur (112–120 °C)[32], there is potential for partial sulfur powder sintering or densification to occur at relatively low temperatures, facilitating the direct lamination of the sulfur powder bed under controlled temperature and pressure conditions. To investigate the thermal characteristics of sulfur, operando X-ray diffraction (XRD) was conducted, showing thermal lattice expansion (diffraction peak shifting to lower angle) above 60 °C (Figs. 1a and S1 and Table S3). Powder rheology tests of elemental sulfur at room temperature (RT, 25 °C) and 80 °C (Fig. 1b and Table S4) revealed increased softening and interparticle bonding of sulfur particles at higher temperatures[33,34]. Scanning electron microscopy (SEM) images (Fig. 1c−f) visualize that as the sulfur powder bed (SEM image of sulfur raw powder is shown in Fig. S2) roll-pressed at various temperatures (RT, 40, 60, and 80 °C). At RT and 40 °C (Fig. 1c, d), the sulfur particles retain their original granular structure with visible inter-particle gaps, indicating minimal particle deformation or contact. As the temperature increases to 60 °C (Fig. 1e), the particles begin to show signs of localized deformation and slightly improved inter-particle contact, though some void spaces persist. Notably, at 80 °C (Fig. 1f), the sulfur powder bed undergoes significant morphological transformation. The particles exhibit pronounced deformation and a substantial reduction in inter-particle voids, resulting in a highly compact and smooth surface texture. This densification indicates a critical threshold where thermal softening, combined with applied pressure, facilitates extensive surface diffusion or solid-state sintering. The substantial contrast between 60 and 80 °C highlights the temperature-dependent consolidation mechanism, suggesting that 80 °C induces sufficient particle plasticity to promote extensive particle reorganization and bonding, even without sulfur melting.

To understand the pronounced morphological transformation observed at 80 °C, the thermal compression behavior of elemental sulfur particles under conditions mimicking the roll-pressing process was simulated. The initial height of the powder bed ($h_i$) of 50 mm, $P = 50$ MPa, and $t = 1.0$ s at RT (25 °C for the simulation) and 80 °C (Fig. 1g–j). The simulation provides insights into particle-level deformation and densification under thermal and mechanical stresses, illustrating how elevated temperatures influence sulfur particle rearrangement, deformation, and packing efficiency. Note that the simulation is designed for qualitative insights into bulk deformation and densification during powder compaction. To balance accuracy and efficiency, we used millimeter-scale particles, which capture bulk mechanical responses acceptable computational cost, whereas a quantitative comparison with experimental data would require micrometer-scale particles that account for adhesive forces. At RT, the particles exhibited elastic and plastic deformation, with a final height ($h_f$) of 24.7 mm, yielding a compaction ratio ($h_i/h_f$) of 2.1. The porosity is 53.2%, estimated as the ratio of empty space to particle volume. At 80 °C, plastic deformation becomes more prominent due to reduced stiffness and increased Poisson's ratio ($v$), resulting in $h_f$ of 15.5 mm, a compaction ratio of 3.1, and porosity of 38.6%. Figure 1j compares the

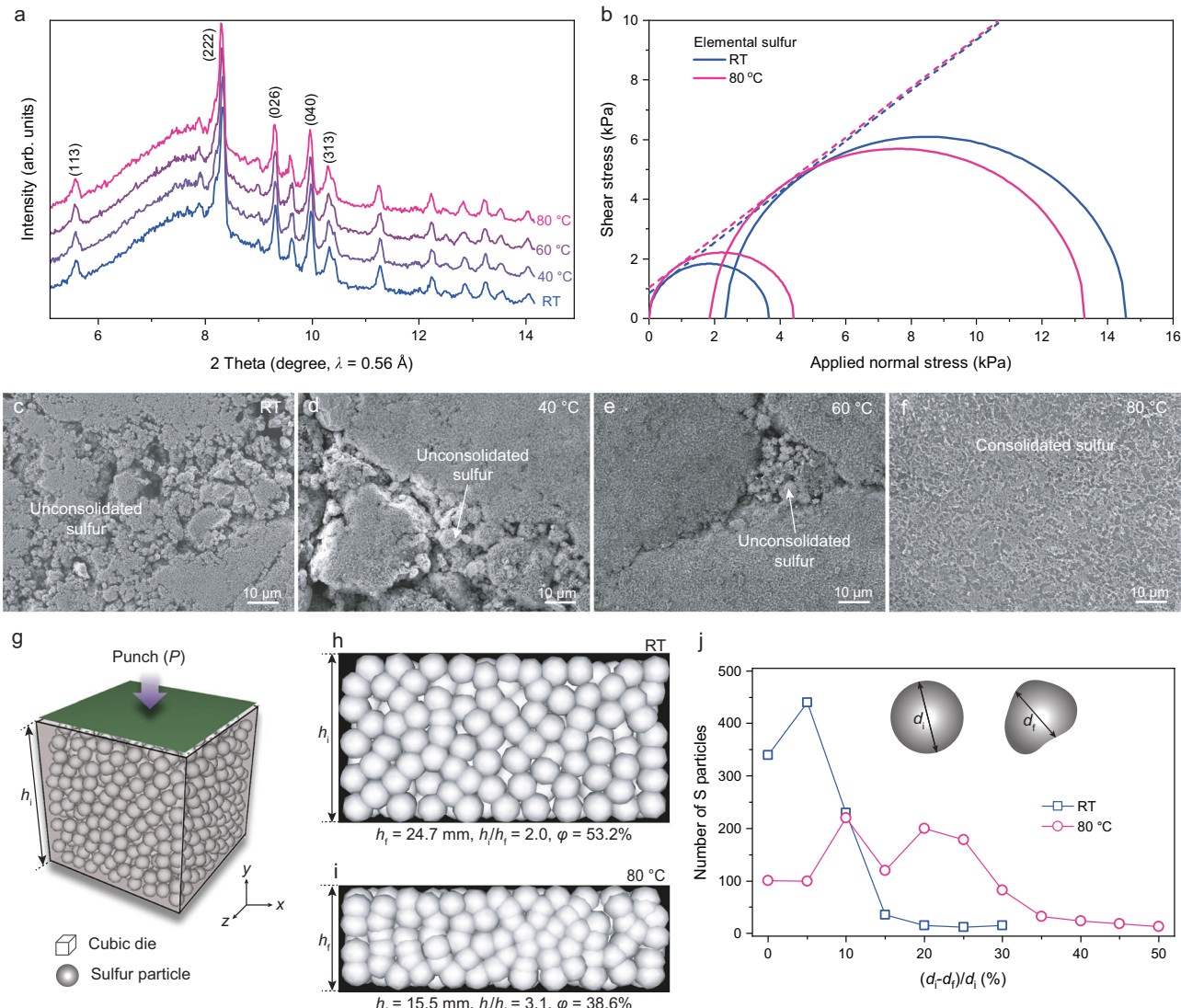

**Fig. 1 | Temperature-dependent structural and mechanical evolution of elemental sulfur. a** Operando XRD patterns of elemental sulfur under heating conditions. The temperature was elevated from room temperature (RT) to 80 °C with a ramping temperature of 1 °C per minute. **b** Mohr circle of elemental sulfur obtained at RT and 80 °C. SEM images of roll-pressed sulfur powder at different temperatures of **c** RT, **d** 40 °C, **e** 60 °C, and **f** 80 °C. Computational modelling of elemental sulfur powder bed compaction behavior at RT and 80 °C. **g** 3D schematic of the computational domain. xy-planar view of the compressed powder bed at **h** RT and **i** 80 °C. **j** Comparison of the number of powder particles as a function of a deformation rate at RT and 80 °C. ($\varphi$, porosity; $h_i$, initial powder bed height; $h_f$, final powder bed height; $d_i$, initial particle diameter; $d_f$, final particle diameter).

number of particles undergoing plastic deformation at RT and 80 °C. The final particle diameter ($d_f$) is calculated from the minor axial length. At RT, most particles showed moderate deformation, with an average $d_f$ of 4.0 ± 0.5 mm and a 5−10% variation in $d_i$ due to higher stiffness at low temperatures. At 80 °C, $d_f$ decreased by 20−30%, averaging 3.0 ± 0.7 mm, confirming that sulfur particles exhibited ductile plastic deformation, forming a dense powder bed at higher temperatures. To achieve similar compactness of sulfur powder bed between RT and 80 °C, approximately four times higher pressure is needed at RT (Fig. S3). This behavior forms the basis of our next section on heat- and pressure-assisted compression of S-C composites, where the role of sulfur in achieving high-density, cohesive electrode structures will be further explored.

### Polymer binder-free S-C composite electrode fabrication

The cohesive nature of sulfur at elevated temperatures, as previously demonstrated, enhances the mechanical integrity of the compressed powder bed. Consequently, when thermally assisted compression

effectively facilitates the distribution of sulfur within the S-C composite powder bed, enabling inter-particle bonding, the fabrication of S-C composite electrodes through solvent-free processes without the need for a polymer binder becomes feasible. Conventional designs of S-C composites generally focus on confining sulfur in nanosized carbon pores to prevent sulfur agglomerates, which helps reduce resistance and mitigate the Li-PS shuttling effect[35]. Our approach, however, combines nano-confined sulfur within porous carbon with micron-sized sulfur particles embedded in S-C secondary particles to facilitate polymer binder-free electrode fabrication. A two-step melt diffusion method was used to synthesize the S-C composite with Ketjen black as the carbon host, and the powder was ground and sieved for uniformity.

SEM (Fig. 2a) with energy-dispersive spectroscopy (EDS) elemental mapping (Fig. S4), bright field transmission electron microscopy (TEM, Fig. S5), high-angle annular dark-field (HAADF, Fig. 2b), and energy loss spectroscopy (EELS) mapping (Fig. 2c) images in the synthesized S-C composites reveal a highly porous morphology,

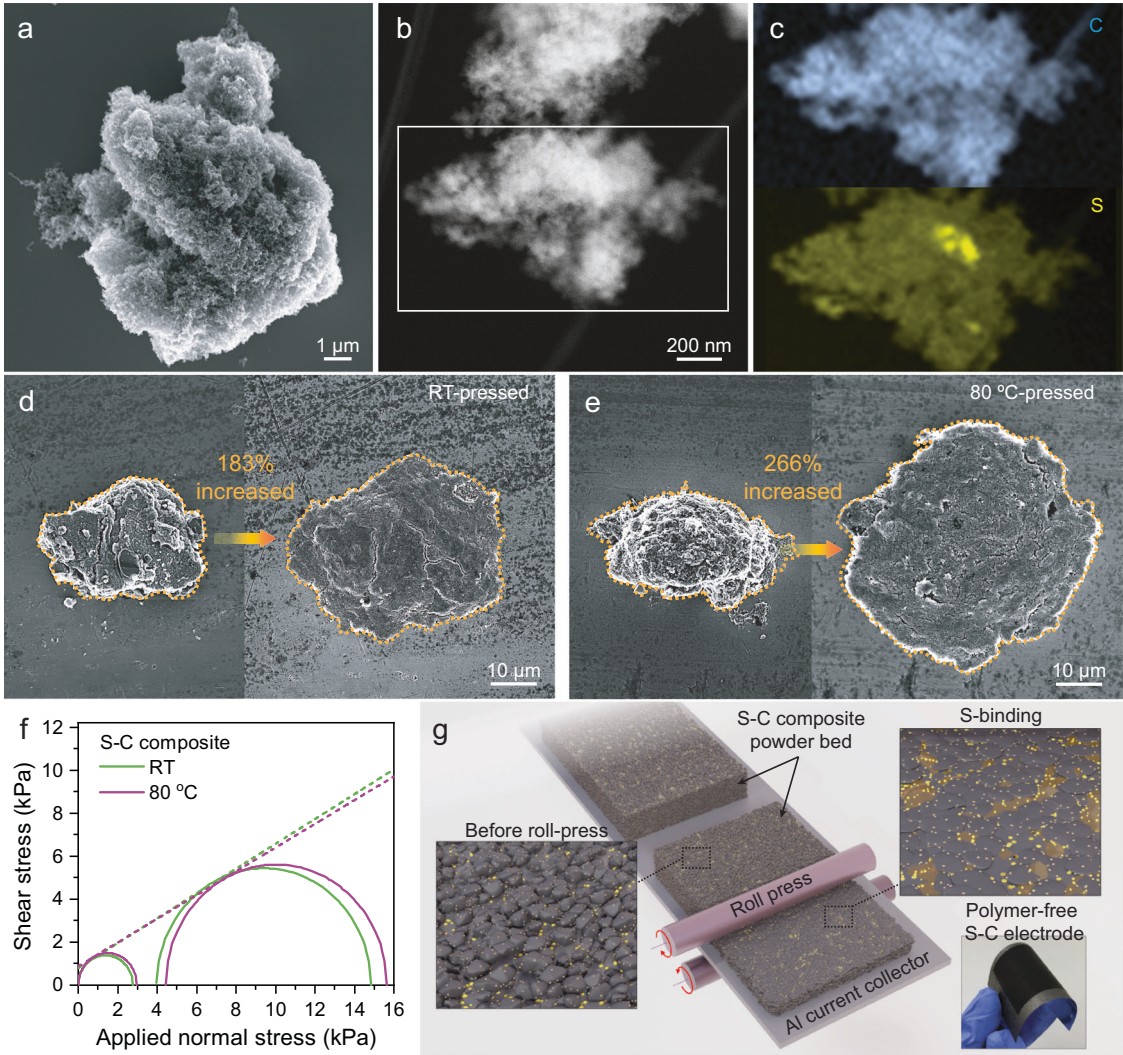

**Fig. 2 | Sulfur-carbon (S-C) composite microstructure and temperature-dependent mechanics enabling binder-free casting. a** scanning electron microscopy image (SEM) of S-C composite powder at low magnification. **b** high-angle annular dark-field scanning transmission electron microscopy images of S-C composite particles and **c** electron energy-loss spectroscopy elemental mapping for carbon and sulfur. SEM images of S-C composite particle before (left) and after (right) compressed at **d** room temperature (RT) and **e** 80 °C. **f** Mohr circle of S-C composite obtained at RT and 80 °C. **g** Schematic illustration of the developed dry and binder-free casting method.

typical of Ketjen black, with sulfur likely confined within the porous structure. The HAADF and EELS image provides further insight into the internal structure of the composite, confirming the hybrid structure where nano-confined sulfur coexists with micron-sized sulfur in S-C composite secondary particles. XRD (Fig. S6) shows crystalline α-sulfur (JCPDS 08-0247) in the composite, and to further investigate local structural features, including non-crystalline sulfur, we performed pair distribution function (PDF) analysis for elemental sulfur and S-C composite (Fig. S7). Compared to elemental sulfur, the S-C composite exhibited broadened features in I(Q) and S(Q)[36,37], along with diminished long-range correlations in G(r)[38,39], indicating the presence of amorphous sulfur domains. The preservation of nearest-neighbor S−S distances further suggests that local sulfur bonding is retained despite structural disorder induced by confinement within the porous carbon matrix. Sulfur content in S-C composite of ~70 wt.% was confirmed by thermogravimetric analysis (TGA, Fig. S8). From the integrated peak areas of XRD patterns shown in Fig. S9, the crystalline sulfur fraction in the S-C composite was estimated to be approximately 44 wt.% of the total composite, corresponding to about 63 % of the total sulfur content. This partial amorphization suggests effective sulfur dispersion and confinement within the carbon matrix. The particle size

distribution and morphology analysis of the S-C composite (Fig. S10 and Table S6) indicates that most of the particles fall within the size range of 30.4 μm and their morphology is relatively well-rounded (circularity of 0.896), which is advantageous for uniform formation of S-C composite powder bed. SEM (Fig. 2d, e) shows a 266% larger flattened area in particles compressed at 80 °C compared to RT (186%), indicating greater compressibility, corresponding to rheological data of S-C composite (Fig. 2f and Table S7), confirming improved cohesion at higher temperatures. Higher-magnification SEM images (Fig. S11) show that RT-pressed samples exhibit uneven sulfur distribution with particles remaining outside aggregates, while 80 °C-pressed samples exhibit more uniform morphology and sulfur distribution.

Using the developed S-C composite containing both nano-confined and micron-sized sulfur, a polymer binder-free S-C electrode was fabricated through a solvent-free manufacturing process. In this approach, the S-C composite powder bed was directly deposited onto a carbon-coated Al foil current collector and laminated via thermally assisted compression. A heat-roll press enables precise pressure control by adjusting the gap, roll temperature, and feeding speed. This process, illustrated in Fig. 2g, uses elevated temperatures to soften sulfur in S-C composites, promoting powder bed consolidation and

inter-particle bonding. Micron-sized sulfur serves as a natural binder, while both micron-sized and nano-confined sulfur function as active materials. This dual role eliminates the need for additional binders or conductive agents, facilitating the formation of a binder-free S-C composite network. The heat-roll pressing of a Ketjen black powder bed under identical compression conditions fails to form a mechanically stable film on the Al current collector (Fig. S12a). In contrast, the sulfur–Ketjen Black mixture formed a coherent film, highlighting the critical role of sulfur as a binder (Fig. S12b).

In this work, positive electrodes with an areal sulfur loading of $3\,mg_S\,cm^{-2}$ and porosity of 51–56% were prepared at 80 °C (referred to as 80 °C-pressed), with another set prepared at RT (referred to as RT-pressed) for comparison (Table S8). Due to the higher stiffness of S-C composite at RT, a narrower roll press gap was used for RT-compression process. The XRD patterns (Fig. 3a) of the RT-pressed and 80 °C-pressed electrodes show structural changes with distinct peak differences, indicating better structural reorganization and compaction at 80 °C. The compression at RT leads to some structural disorder of α-sulfur phase, as evidenced by peak broadening, but lacks the more pronounced crystallographic changes (i.e., peak shifting) seen at 80 °C. The shift and associated peak shape distortion of α-sulfur in 80 °C-pressed electrodes suggest thermomechanical strain or local lattice distortion within α-sulfur, likely facilitating improved particle–particle cohesion and mechanical integrity in the composite electrode. Peel force measurements (Fig. S13) confirm that 80 °C-pressed electrodes exhibit enhanced mechanical stability over the RT-pressed electrodes.

To gain deeper insight into the process, a structural analysis of the RT-pressed and 80 °C-pressed electrodes was performed using X-ray micro-computed tomography (μ-CT), SEM, and OM with surface profilometry. Despite the fact that RT-pressed and 80 °C-pressed electrodes have approximately the same bulk porosity, the X-ray μ-CT results in Fig. 3b (RT) and 3e (80 °C) reveal key differences: the RT-pressed electrode shows a non-uniform void structure with more complex pore pathways (tortuosity: 3.14), while the 80 °C-pressed electrode has a more uniform structure with direct pore pathways (tortuosity: 2.79). The reduction in tortuosity is expected to facilitate more efficient Li-ion migration by shortening diffusion paths, which can contribute to improved rate capability and electrochemical uniformity during cycling. Top and cross-sectional view SEM images and porosity distribution analysis by X-ray μ-CT of the RT-pressed (Fig. 3c, d) and 80 °C-pressed electrodes (Fig. 3f, g) indicate uniform porosity for the 80 °C-pressed electrode, while the RT-pressed electrode has lower porosity at the top and bottom. SEM and OM with surface profilometry results (Figs. 3c, f and S14 and Table S8) reveal the 80 °C-pressed electrode has a higher surface void fraction (49%) and greater roughness ($S_{pc}$ of $1612.7\,mm^{-1}$) compared to the RT-pressed electrode (40% void fraction, $S_{pc}$ of $1192.0\,mm^{-1}$). Although the S-C composite deforms and compresses more effectively at 80 °C than at RT, the 80 °C-pressed electrode exhibits a rougher surface, interestingly. This difference arises from how pressure is distributed to achieve the same bulk porosity (51–56%) at varying temperatures. At RT, higher pressure was given to the powder bed to address the higher stiffness of the S-C composite particles, resulting in higher pressure concentrated at the top and bottom of the powder bed. This uneven pressure leads to smoother surfaces but non-uniform porous structures, as seen in cross-sectional images and X-ray μ-CT. In contrast, at 80 °C, the S-C composite compresses more uniformly due to increased compressibility, requiring lower pressure for the same porosity, creating a rougher surface and more consistent porous structure. This effect becomes more pronounced as the areal mass loading increases from 1 to $5\,mg_S\,cm^{-2}$ (Table S8 and Figs. S15–S17). At low loadings, pressure transfers more uniformly throughout the powder bed, but with higher loadings, thicker beds show greater inhomogeneity, especially at RT, leading to a more non-uniform structure.

X-ray μ-CT analysis results of the slurry-cast S-C composite electrodes (Fig. S18) reveal significant differences between binder-based and binder-free S-C composite electrodes, particularly in terms of tortuosity and structural integrity. The slurry-cast electrodes display a more tortuous pore network with pronounced microscale defects than the RT and 80 °C-pressed electrodes, as evident from the top-view SEM and OM and surface topology analysis (Fig. S19 and Table S9) and the 3D reconstructed X-ray μ-CT images (Fig. S18). High-speed camera observations reveal that the electrolyte absorptivity of the electrodes differ significantly between the binder-free and slurry-cast S-C composite electrodes. The binder-free electrodes display more rapid and uniform electrolyte absorption, with contact angles of 17° and 14° after 0.02 s for RT-pressed and 80 °C-pressed samples, respectively, compared to 23° for the slurry-cast S-C composite electrode. This enhanced wettability is attributed to their more homogeneous porosity and structural integrity (Fig. S20). This faster absorptivity ensures improved wetting of the electrode and enhanced ion transport pathways, which could contribute to the enhancement in the electrochemical performance of the binder-free electrodes. In contrast, the binder-containing slurry-cast electrodes, with their more tortuous pore network and the presence of a polymer binder, exhibit slower and less uniform electrolyte absorption, leading to poorer wetting. After 2 s, the RT-pressed electrode shows surface swelling and irregularity (Fig. S20), indicating localized expansion and reduced mechanical stability. In contrast, the 80 °C-pressed and slurry-cast electrodes maintain flat, intact surfaces, suggesting better structural resilience. These findings, in conjunction with the X-ray μ-CT and SEM observations, emphasize the critical role that microstructural uniformity and electrolyte accessibility play in the overall performance of S-C composite electrodes.

## Electrochemical tests

The electrochemical performance of the S-C composite positive electrodes ($3\,mg_S\,cm^{-2}$) was systematically evaluated through a series of tests, including rate capability, cyclic voltammetry (CV), electrochemical impedance spectroscopy (EIS), and galvanostatic charge-discharge cycling. While pouch cells are important for assessing practical energy metrics[40–42], coin cells were used to enable controlled evaluation of the intrinsic properties of the sulfur positive electrodes developed in this work. The rate capability test (Figs. 4a and S21) demonstrates that the 80 °C-pressed electrodes exhibit higher rate performance across a wide range of C-rates with specific capacities of approximately from 1300 to 600 mAh $g^{-1}$ from 0.05 to 1.0 C, respectively, whereas the RT-pressed electrodes (70 wt.% sulfur) show a dramatic reduction in capacity as the C-rate increases, particularly above 0.5 C rate. The cycling tests (Fig. 4b) validate these improvements, with the 80 °C-pressed electrodes retaining significantly more capacity over 200 cycles compared to the RT-pressed electrodes, which decayed after 60 cycles. The slurry-cast electrode containing a polymer binder prepared for comparison ($3\,mg_S\,cm^{-2}$, 58 wt.% sulfur) showed even worse cycle performance and rate performance with rapid capacity decay and early failure than both binder-free positive electrodes. The inset chart in Fig. 4b shows that the slurry-cast electrode exhibited a specific capacity of only 627 mAh $g_{electrode}^{-1}$ (based on the mass of the electrode film vs. 896 mAh $g_{electrode}^{-1}$ for 80 °C-pressed electrodes) during the initial cycle. This performance declined substantially by the 90th cycle, decreasing to 320 mAh $g_{electrode}^{-1}$. A gradual decline in Coulombic efficiency was observed during extended cycling (Fig. 4b), particularly in the RT-pressed and slurry-cast samples, indicating increasing interfacial resistance. Figure 4c−e highlight the more stable voltage profile of the 80 °C-pressed electrodes with lower charge and discharge overpotentials, compared to the RT-pressed electrodes. The CV results (Fig. 4f−h) align with previous findings, showing sharper and more distinct redox peaks for the 80 °C-pressed electrodes, indicating better electrochemical reversibility and more

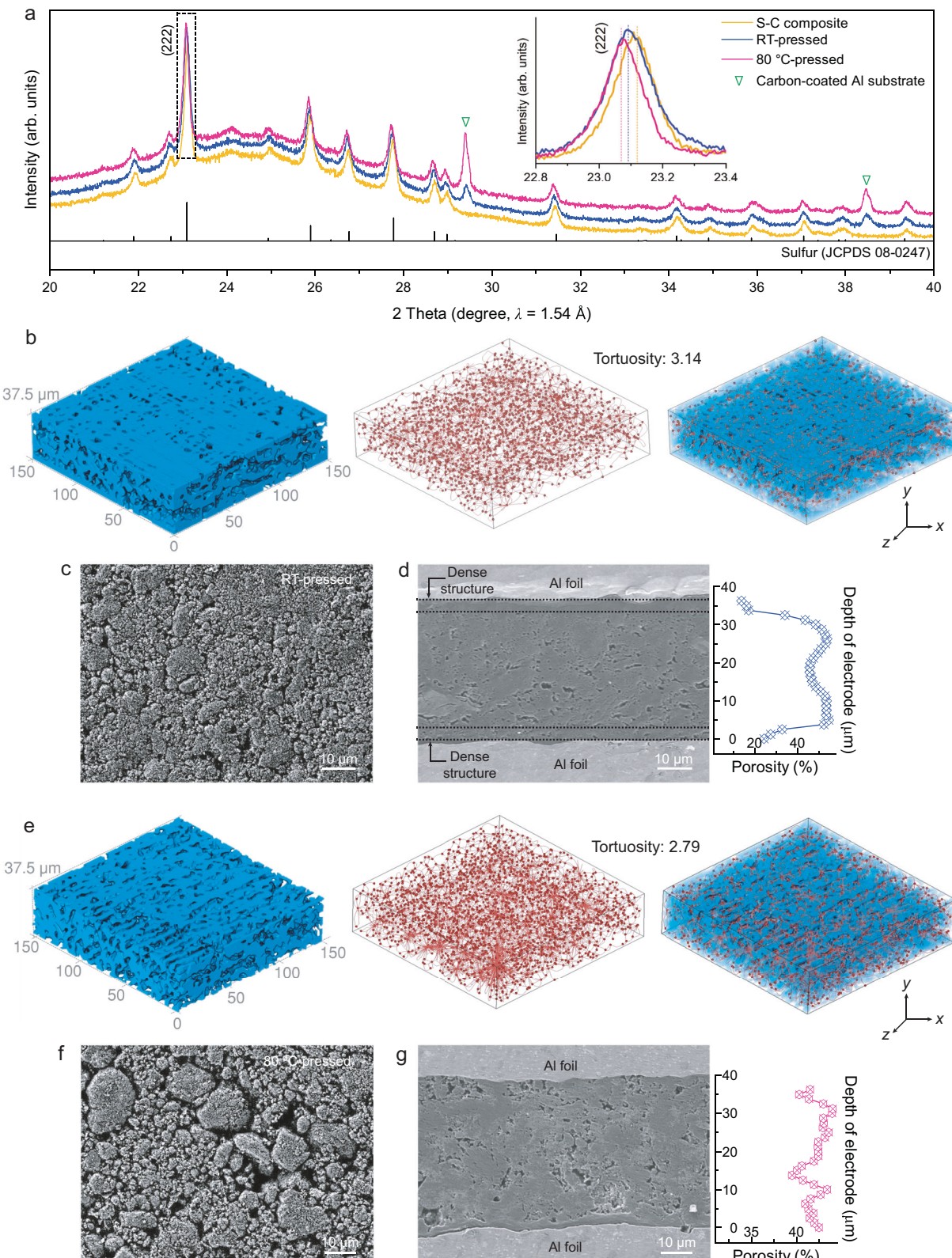

**Fig. 3 | Multimodal characterization of binder-free sulfur-carbon (S-C) composite electrodes prepared at room temperature (RT) and 80 °C. a** XRD patterns of the S-C composite powder and binder-free S-C composite positive electrodes with aluminum (Al) foil substrate prepared via the solvent-free process at RT and 80 °C. **b** X-ray micro computed tomography (μ-CT) analysis and **c** top-view SEM, and **d** cross-sectional SEM images of the RT-pressed electrodes, **e** X-ray μ-CT analysis and **f** top-view SEM, and **g** cross-sectional SEM images of the 80 °C-pressed electrodes. Porosity distribution along the electrode thickness, as revealed by X-ray μ-CT, is shown adjacent to the cross-sectional SEM images.

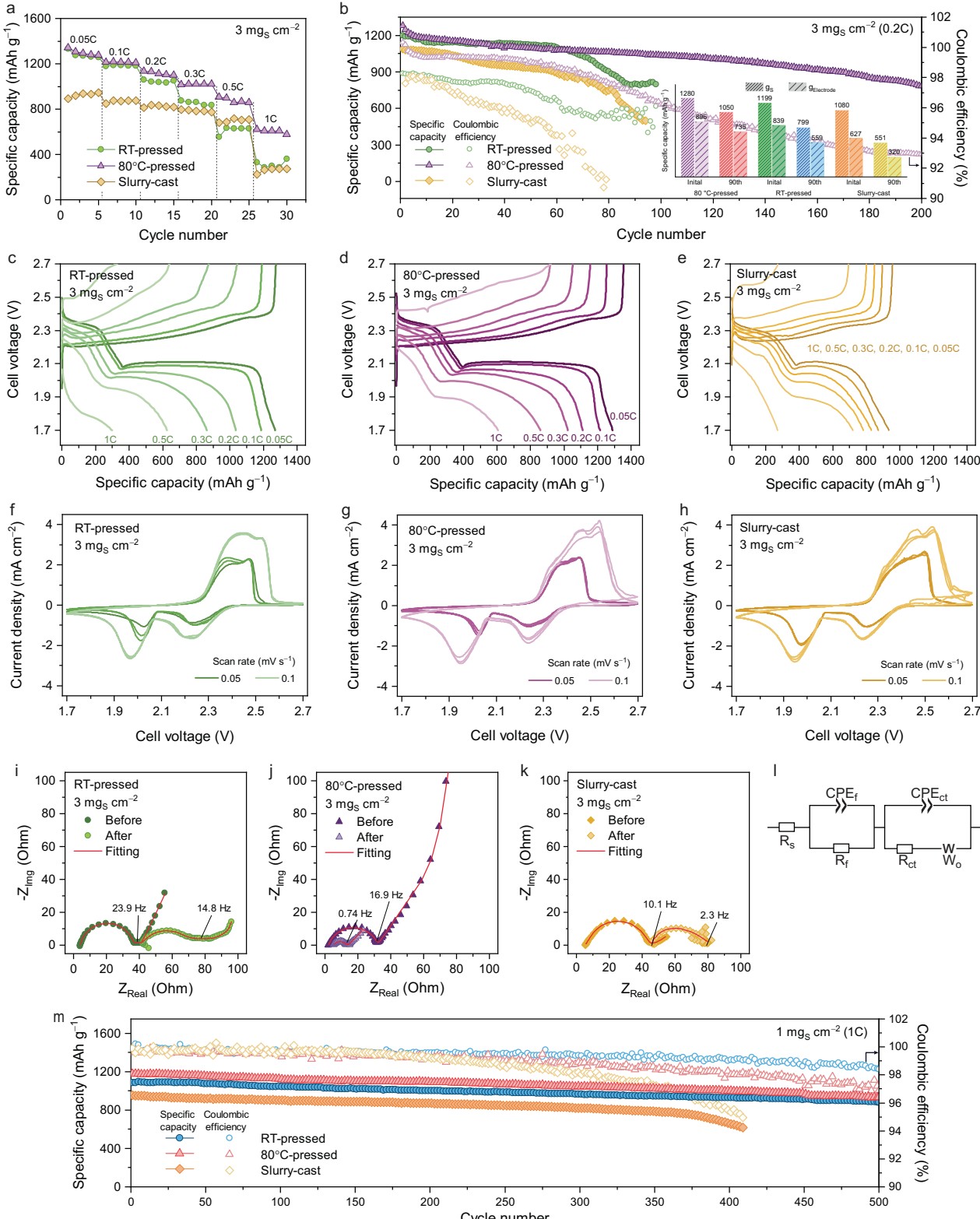

**Fig. 4 | Electrochemical characterization of sulfur-carbon (S-C) composite positive electrodes (1 C = 1675 mA g$_S^{-1}$, testing temperature: 25 °C). a** Rate capability and **b** galvanostatic charge-discharge cycling at 0.2 C for the room temperature (RT)-pressed and 80 °C-pressed, and slurry-cast S-C composite positive electrodes (areal sulfur loading: 3 mg$_S$ cm$^{-2}$). Voltage profiles of **c** RT-pressed, **d** 80 °C-pressed, and **e** slurry-cast electrodes at various C-rate. Cyclic voltammetry (CV) curves of the **f** RT-pressed, **g** 80 °C-pressed **h** slurry-cast electrodes (areal

sulfur loading: 3 mg$_S$ cm$^{-2}$). Nyquist plots of **i** RT-pressed, **j** 80 °C-pressed, and **k** slurry-cast electrodes measured at the fully charged state before and after 30 cycles, with **l**) the corresponding equivalent circuit model (areal sulfur loading: 3 mg$_S$ cm$^{-2}$, (CPE, constant phase element; $R_s$, solution/electrolyte resistance; $R_f$, film resistance; $R_{ct}$, charge-transfer resistance; $W_o$, finite-length Warburg element). **m** Galvanostatic cycling at 1.0 C for the RT- and 80 °C-pressed, and slurry-cast electrodes (areal sulfur loading: 1 mg$_S$ cm$^{-2}$).

efficient sulfur redox reactions. EIS analysis performed at the charged state after 30 cycles (Fig. 4i–k and Table S10) revealed distinct trends for each electrode type. The 80 °C-pressed electrodes maintained low impedance, while both the RT-pressed and slurry-cast electrodes exhibited showed clear increases in ohmic resistance ($R_s$). The rise in $R_s$ for the RT-pressed electrode suggests compromised structural integrity, likely due to mechanical degradation during cycling[43–45], whereas the thermally compressed 80 °C-pressed electrode retained a stable architecture. Similarly, the increase in $R_s$ for the slurry-cast electrode is consistent with previous reports that conventional slurry processing can lead to solvent-induced inhomogeneity and poor interfacial stability issues[46–48]. The 80 °C-pressed electrodes was further evaluated in a single-layer pouch cell ($1.5 \times 1.5$ cm$^2$, 3 mg$_S$ cm$^{-2}$, Fig. S22), which delivered an initial capacity of 1000 mAh g$^{-1}$ and maintained 897 mAh g$^{-1}$ after 30 cycles, demonstrating the applicability of the binder-free S-C electrodes to a more practical cell form factor and consistent electrochemical behavior with the corresponding coin-cell results.

In the cycling tests (Fig. 4l) conducted at a lower mass loading of 1 mg$_S$ cm$^{-2}$, both the binder-free positive electrodes demonstrated stable cyclability at 1 C. However, the 80 °C-pressed electrodes sustained a higher capacity of approximately 1000 mAh g$^{-1}$ over 500 cycles. In contrast, the slurry-cast positive electrode failed after 400 cycles, despite showing slightly low charge and discharge overpotentials across various C-rates, comparable to those of the binder-free positive electrodes (Fig. S23). The improved performance of the binder-free positive electrodes highlights the limitations of conventional slurry casting processes in achieving the necessary structural and electrochemical stability, as demonstrated by the underperformance of the slurry-cast S-C composite positive electrodes compared to both the RT- and 80 °C-pressed electrodes. This is also featured at a higher mass loading of 5 mg$_S$ cm$^{-2}$ (Fig. S24) as well. While consistently delivering higher capacities across all C-rates in the rate performance test and at 0.1 °C during cycling, the 80 °C-pressed electrodes (5 mg$_S$ cm$^{-2}$) exhibited Coulombic efficiency values slightly above 100%, likely due to delayed activation of isolated sulfur domains within the sulfur positive electrodes, unlike the consistently lower Coulombic efficiency observed in the RT-pressed and slurry-cast S-C positive electrodes. EIS analysis for the 1 and 5 mg$_S$ cm$^{-2}$ electrodes revealed trends consistent with those observed at 3 mg$_S$ cm$^{-2}$, reinforcing the generality of the impedance behavior across various loadings (Figs. S23 and S24 and Table S10). The electrochemical analyses reveal that the S-C composite positive electrodes pressed at 80 °C exhibit significantly enhanced performance compared to those pressed at RT. This improvement can be directly attributed to the optimized microstructure achieved during compression at elevated temperatures, as demonstrated in the previous characterization results. The more uniform porosity distribution, low tortuosity, and enhanced structural integrity of the 80 °C-pressed electrodes facilitate efficient ion transport and electron conduction, resulting in reduced polarization and improved electrochemical kinetics[49,50]. In contrast, the RT-pressed electrodes suffer from inhomogeneous compression and lower porosity at the electrode surface, which impedes their overall performance. These findings are further supported by the structural differences observed, where the more uniform compression of the 80 °C-pressed electrodes leads to better contact between the active material and the conductive matrix. The enhanced structural and compositional uniformity at higher processing temperatures results in lower interfacial resistance, attributed to improved adhesion between the active materials and the current collector, as well as better particle-to-particle contact within the positive electrode. This uniform microstructure contributes to improved cycling stability, minimizing electrode degradation, sulfur loss, and active material disconnection during repeated charge-discharge cycles. In contrast, the RT-pressed electrodes exhibit faster capacity fading due to their less stable and inhomogeneous structure.

## In situ and ex situ characterizations

In situ OM conducted during the initial discharge provides complementary insights into the dynamic behavior of the positive electrodes (Fig. 5 and Supplemental Movies 1 and 2). At the onset of lithiation, both the RT-pressed and 80 °C-pressed electrodes exhibit the formation of Li-PS, indicated by the color change of the electrolyte to light orange. As discharge progresses, both positive electrodes undergo volume expansion due to sulfur lithiation, accompanied by a darkening of the electrolyte color, reflecting an increased concentration of Li-PS[51]. Due to the spacer-free design of the in situ OM cell, the larger interelectrode gap results in higher polarization than in conventional coin cells. While no evidence of particle detachment was observed, the volume expansion behavior differed significantly between the two electrodes. During the initial 10% depth of discharge (DOD), the RT-pressed electrodes exhibit localized and uneven volume changes, with specific regions of the electrode expanding disproportionately, resulting in uneven mechanical deformation (Fig. 5b). In contrast, the 80 °C-pressed electrodes display a more uniform volume expansion across the electrode, consistent with improved structural cohesion and more uniform sulfur distribution in the 80 °C-pressed electrodes. Comparative OM images during both discharge and charge at various DOD and states of charge, as shown in Fig. S25, further clarify these differences. The fully charged positive electrode pressed at 80 °C exhibits a uniform reformation of sulfur from Li-PS, returning to its original microstructure. Conversely, the RT-pressed electrode shows a sulfur structure largely confined within the carbon matrix, resulting from microstructural deformation.

Ex situ SEM images of positive electrodes after cycling (Fig. 5c, d) further confirm these observations, revealing clear differences in the mechanical stability and surface morphology of the RT- and 80 °C-pressed samples. The RT-pressed electrodes display significant surface cracking and delamination after 5 cycles, indicating mechanical degradation caused by repeated volume expansion and contraction during Li-S redox reactions. Conversely, the 80 °C-pressed electrodes maintain a crack-free surface, with no signs of delamination or particle separation, even after the same number of cycles. Additional ex situ SEM images collected at ~40% DOD at the initial discharge process reveal that the 80 °C-pressed electrodes retain structural cohesion during early electrochemical cycling (Fig. S26). This observation, together with prior findings that sulfur conversion does not occur uniformly across the electrode[52], suggests that partially unreacted sulfur and early-stage Li$_2$S$_x$ species can coexist, allowing localized structural retention during the formation of discharge products. Importantly, the observations suggest that the initial mechanical integrity provided by heat-pressing plays a critical role in mitigating the potential structural weakening associated with the solid–liquid–solid transition of sulfur during cycling. Although polysulfide formation and subsequent sulfur precipitation inherently involve relocation of active material, the 80 °C-pressed electrode preserves a cohesive framework that accommodates these transformations without particle detachment or severe cracking, as confirmed by ex situ SEM. S||Li$_2$S cell testing (Fig. S27) in which a lithiated positive electrode (Li$_2$S-containing) was paired with a fresh S/C positive electrode further confirmed enhanced reversibility in 80 °C-pressed electrodes, which exhibited lower charge polarization and more stable cycling behavior compared to RT-pressed samples, consistent with improved structural integrity and Li$_2$S distribution. This indicates that effective thermomechanical compression, by promoting structural uniformity, not only ensures intimate interparticle contact at the outset but also helps preserve the electrode architecture throughout repeated Li-S redox cycles, as summarized in Fig. 5e, f. These collective findings of ex situ SEM and S||Li$_2$S symmetric cell test highlight that the

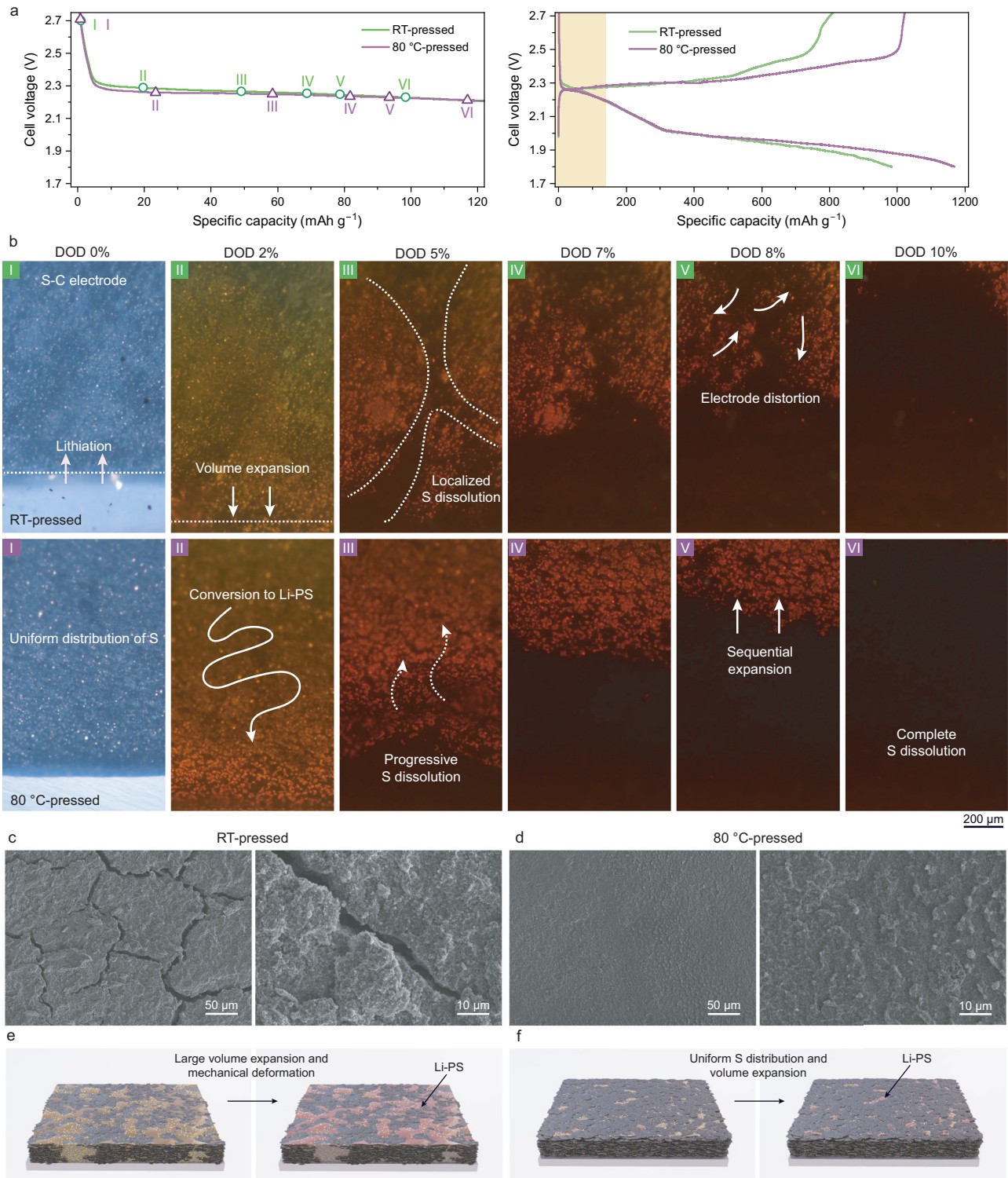

**Fig. 5 | Microstructural deformation of room temperature (RT)-pressed and 80 °C-pressed sulfur-carbon (S-C) composite electrodes during electrochemical cycling (areal sulfur loading: 3 mg$_S$ cm$^{-2}$, 1 C = 1675 mA g$_S^{-1}$, testing temperature: 25 °C). a** Voltage profiles of the initial charge-discharge processes at 0.1 °C within a cell voltage range of 1.8–2.7 V at 25 °C during the in situ optical microscopy (OM) experiment, with an enlarged view of the first 10% Depth of discharge (DOD) highlighted (yellow). **b** OM images captured during the first discharge at DOD 0, 2, 5, 7, 8, and 10%. Ex situ scanning electron microscopy (SEM) images of **c** the RT-pressed and **d** 80 °C-pressed positive electrodes after five cycles at 0.2 °C, highlighting structural differences. Schematic illustrations depicting the structure of **e** the RT-pressed and **f** 80 °C-pressed positive electrodes before and after cycling.

improved morphological integrity of the 80 °C-pressed electrodes not only supports structural robustness but also accommodates the chemo-mechanical stresses induced by repeated sulfur redox cycling. Given that sulfur serves both as the active material and a structural binder in this solvent- and binder-free system, its homogeneous distribution within the carbon matrix is particularly critical. The resulting uniform composite architecture enhances interparticle cohesion and mitigates fracture propagation during solid–liquid–solid transitions.

This structural coherence is essential to maintaining continuous electron and ion transport pathways, preventing the detachment of active material, and ultimately enabling prolonged cycling stability. The findings presented here provide valuable insights into the relationship between processing conditions, structural integrity, and electrochemical performance, offering a practical framework for designing next-generation sulfur-based energy storage systems.

## Methods

### Characterization of elemental sulfur
Operando XRD measurements during heating were performed using a STADI P STOE dual-transmission diffractometer equipped with an Ag Kα source ($\lambda = 0.55941$ Å, 40 kV, 40 mA). Data were collected with a step size of 0.015° and a dwell time of 15.81 s per step to investigate the thermal behavior of elemental sulfur (sulfur nanopowder, 99.99%, Skyspring). At each temperature, diffraction patterns were acquired for 10 min total, consisting of two 5 min scans at successive detector positions. To investigate particle deformation and densification, a powder bed of elemental sulfur was deposited onto Al current collectors and compressed using a heat-roll press (Wellcos, WCRP-1015HG) at various temperatures. The compression pressure applied to the sulfur powder bed was kept consistent across all samples by maintaining a fixed gap between the rolls and the roll rotation speed. SEM analysis (Auriga Zeiss) was performed on the compressed powder bed to examine the structural changes resulting from the thermal and mechanical treatments. Powder rheological properties were evaluated using a powder shear cell tester (TA Instruments, Discovery HR-20). Over 20 mL of powder was loaded into a stainless steel cylindrical cell (28 mm diameter). The assembly was then heated to the target temperature—either RT or 80 °C—at a ramp rate of 3 °C min⁻¹. Upon reaching the set temperature, the powder was equilibrated for 5 min to ensure thermal uniformity throughout the sample. A pre-consolidation pressure of 9 kPa was applied to condition the powder bed and achieve a uniform packing state. Excess powder was removed after consolidation, resulting in an exact sample volume of 18 mL for shear testing. The sample was then subjected to a constant shear rate ($0.001$ rad s⁻¹) under fixed normal stress until shear failure occurred[53]. The normal stress was systematically decreased to 7, 6, 5, 4, and 3 kPa in successive cycles, following instrument protocols. A yield locus was constructed by plotting the steady-state shear stress versus normal stress, and a linear fit was applied. Using Mohr's circle analysis, the unconfined yield strength (UYS) and the major principal stress (MPS) were extracted: (1) The first Mohr's circle was constructed with one point at the origin (0,0) and a second point tangent to the yield locus; its x-axis intersection yields the UYS. (2) The second Mohr's circle used the pre-shear point and a tangent to the yield locus; its x-axis intersection defines the MPS[54–56]. The flow function (FF) of the powder was calculated using the UYS and MPS values, as expressed in Eq. (1) shown below.

$$FF = \frac{MPS}{UYS} \qquad (1)$$

### Computation modeling for thermal compression behavior of elemental sulfur
The roll-pressing of sulfur powder bed is modeled using ANSYS Workbench, a commercial software package that integrates ROCKY and Mechanical modules, which implements both discrete element model (DEM) and finite element model (FEM). The objective of the model is to qualitatively understand how temperature changes in the roll-pressing process affect the sizes and shapes of particles in the resulting hot-rolled sintered film. To simplify the modeling process, the actual roll-pressing process was represented as the compaction of sulfur particles in a die compressed by a rigid punch. First, DEM was

used to randomly pack sulfur particles in the simulation domain, which was then imported into the FEM for the compaction simulation. The computational domain, which measures $50 \times 50 \times 50$ mm³, contains polygonal sulfur powder particles modeled as sphero-polygons with 12 vertices, an initial equivalent sphere diameter, $d_i = 5$ mm, and vertical/horizontal aspect ratio of 1. This sphero-polygon representation captures the irregular, faceted geometry of actual particles while enabling computation of inter-particle contact mechanics and packing behavior. The simulation domain was sized to include at least ten sulfur particles with a representative diameter of ~5 mm in each spatial direction, ensuring statistically meaningful assessment of bulk powder behavior. The initial height of the powder bed, $h_i$, is 50 mm. Each powder particle is independent and deformable, with a friction coefficient of 0.2, determined by the Coulomb friction model. The contact between neighboring particles is modeled by the Bond model, similar to the works of Gimenez et al.[57] and Potyondy & Cundall[58]. In the Bond model, a massless cylindrical bond entity is attached to a pair of neighboring particles, exerting elastic and viscous forces and moments on the particles as a reaction to deformations caused by their relative motion. An external load higher than the specified tensile and shear strengths may break the bonds. The bond model parameters used in the simulation include a maximum elongation of 1 mm, a damping ratio of 0.25, a distance factor of 0.1, normal and tangential stiffness per unit area of 100 kN m⁻³, tensile and shear stress limits of 10 kPa, and a bond activation time of 0.1 s. These values were selected to approximate the mechanical behavior of loosely agglomerated particles and to enable realistic fracture and rearrangement during compaction. These Bond model parameters were chosen to represent the general inter-particle mechanics of ductile porous powders, but since they are not material-specific, they cannot be used for direct quantitative comparison with experimental data. In the FEM, a free-fall displacement boundary condition is imposed on the rigid punch, along with an additional force factor that controls the applied pressure, P. As the punch moves downward, it interacts with the particles in the top-most layer, transmitting forces downward to neighboring particles in each layer. As the powder particles are compressed and undergo elasto-plastic deformation, the punch monotonously moves downward for the prescribed punch time, resulting in a final powder bed height, hf. To simulate the powder compaction at RT (25 °C) and 80 °C, the appropriate mechanical properties, i.e., Young's modulus, E, and Poisson's ratio, v, are defined. Due to the limited data for sulfur particles at various temperatures, we assume E = 10 GPa[59] and v = 0.2 at 25 °C. Most crystalline materials possess v in the range of 0.2–0.3 at 25 °C[60]. At 80 °C, we assume E = 5 GPa and v = 0.3, based on the linear model for the temperature dependence of Young's modulus[61]. During pressing, relative density, defined as the volume ratio of particles to the die, varies across three stages. As the punch starts to press, particles rearrange their positions to fill large gaps, resulting in rapid increase in relative density. Next, the positions of particles become interlocked, and the rate of increase in relative density slows down. The particles start undergoing plastic deformation, and the gaps are filled by large-scale plastic deformations. Finally, a dense state is formed, and the relative density no longer increases. Among the three stages, plastic deformation is most influenced by the temperature change.

### Synthesis and characterization of S-C composites
The S-C composite was synthesized via a conventional melt diffusion process. Initially, sulfur nanopowder and Ketjen black EC-600JD (MSE Supplies) were combined in a 75:25 weight ratio and mixed using a mortar and pestle. The S-C mixture was transferred to an alumina crucible and placed in a tube furnace (Lindberg/Blue M Mini-Mite, Thermo) to heat-treat the mixture at 155 °C for 10 h under an argon (Ar) atmosphere to enable the melt diffusion of sulfur into the porous carbon matrix. After natural cooling, the resulting S-C composite was

ground using a mortar and pestle, reloaded into the crucible, and subjected to a second heat treatment at 155 °C for 5 h under Ar atmosphere. After cooling, the final composite was retrieved from the crucible and passed through a 500-mesh (30 μm) ultrasonic sieve (UP200st, Hielscher) operating at 40 W to ensure uniform particle size distribution. The sulfur content in the S-C composite was 70 wt.%, confirmed by TGA

The morphology of the synthesized S-C composite was analyzed using SEM (Auriga Zeiss) with an electron acceleration voltage of 5 keV and a working distance of 5 mm, along with EDS elemental mapping (Oxford Instrument) and TEM (ARM200F, JEOL), accompanied by scanning TEM (STEM) with EDS elemental mapping (Oxford Instrument) and HAADF imaging. Sulfur content in the S-C composite was determined by TGA (TG92 Setaram) at the ramping rate of 5 °C min$^{-1}$ under Helium gas. The crystal structure of the S-C composite was scanned between 10 and 70 2θ degree using Cu-Kα source XRD (PANalytical Aeris Malvern, λ = 1.54060 Å, 40 kV, 15 mA) with a step size of 0.01° and a dwell time of 200 s per step. PDF measurements were performed using a STOE STADI 60 P transmission X-ray diffractometer equipped with an Ag Kα$_1$ source (λ = 0.55941 Å; 40 kV, 40 mA) and a curved Ge(111) monochromator. Diffraction patterns were collected in Debye–Scherrer geometry using four stationary MYTHEN2 R 1 K silicon strip detectors covering a 2θ range of 2°–140° (Q$_{max}$ = 21 Å$^{-1}$). Samples were sealed in spinning borosilicate glass capillaries (0.50 mm outer diameter, 10 μm wall thickness) to minimize preferred orientation and enhance statistical averaging. Each sample was measured for 24 h, and background scattering was subtracted using data from an identically measured empty capillary. The 1D diffraction patterns were processed with PDFgetX3 to yield the background-subtracted intensity I(Q), structure function S(Q), reduced function F(Q), and real-space PDF G(r)[39,62]. To evaluate the amorphous sulfur content in the S-C composites, a calibration series of crystalline S-C (S$_{cryst}$-C) samples was prepared by mechanically mixing sulfur and Ketjen black at weight ratios of 90:10 (90S$_{cryst}$), 70:30 (70S$_{cryst}$), 50:50 (50S$_{cryst}$), and 40:60 (40S$_{cryst}$) without any heat treatment. All S$_{cryst}$-C mixtures were pre-weighed to maintain a constant carbon mass on the sample holder. For reference, pure crystalline sulfur (100S$_{cryst}$), Ketjen black, and the synthesized S-C composites were also analyzed. XRD was performed using a PANalytical Aeris diffractometer with a Cu-Kα source (λ = 1.5406 Å), scanning from 20° to 33° 2θ with a step size of 0.01° and a dwell time of 200 s per step. The integrated intensities of the sulfur (222), (026), and (040) reflections were obtained and used to construct calibration curves of integrated peak area versus crystalline sulfur content. Linear regression yielded R$^2$ values higher than 0.95 for all peaks, indicating strong linearity. The relative crystalline sulfur content of the S-C composite was then determined by extrapolation from the regression model, and the amorphous sulfur fraction was estimated by subtracting the crystalline contribution from the total sulfur content. This analytical approach follows methodologies established in prior XRD-based phase quantification studies[63,64].

Particle size and morphology analysis were done by particle size analyzer (Morphologi G3SE Malvern) at the magnification of ×20 and ×50. To observe the particle deformation behaviors of the S-C composite at different temperatures, the S-C composite particles were dispersed onto poly(vinylidene fluoride)-coated Al foil. First, a PVDF-coated Al foil was prepared by casting a PVDF (Kureha 9300; Molecular weight: > 1.0 × 10$^6$ g mol$^{-1}$) solution in NMP (50 mg mL$^{-1}$). The solution was cast onto Al foil (Wellcos, thickness: 20 μm) with a blade gap of 100 μm, followed by partial drying process in a fume hood at 25 °C for 3 h. Before the solution is completely dried. Subsequently, the S-C composite powder was uniformly dispersed onto the partially dried PVDF film. The samples were then dried overnight in the fume hood, followed by vacuum drying at 50 °C for 12 h. SEM images were acquired prior to calendaring. Subsequent to the initial SEM imaging, the S-C composite on the Al foil underwent calendaring processes at both RT and 80 °C, with the heat-roll press gap remaining consistent in both instances. The compressed particles were subsequently subjected to SEM imaging once again to analyze their deformation characteristics resulting from the calendaring process. The same shear cell-based method described above was employed to characterize the rheological behavior of the S-C composite powder under identical testing conditions.

## Solvent- and binder-free S-C composite positive electrode preparation and characterization

S-C composite powder was carefully loaded into a stainless steel window measuring 4 × 5 cm², which is positioned on carbon-coated Al foil (Wellcos, thickness: 20 μm) to create a 4 × 5 cm² powder bed. The heat-roll press was meticulously preheated to the specific target temperature for a duration of 10 min to ensure consistent and uniform heating throughout the material. The gap between the rolls was then precisely adjusted to align with the desired areal mass loading and processing temperature, guaranteeing the maintenance of a constant bulk porosity. The roller speed was carefully set at 10 cm min$^{-1}$ to achieve the desired lamination process. Following the lamination, electrodes were accurately punched into circular discs using a hand operated punching tool (WC-H125), preparing them for subsequent electrochemical testing within 2032 coin cells (Wellcos). A simple mixture of S (70 wt.%) and Ketjen black (30 wt.%) using a mortar and pestle method was also prepared to compare the physical properties of electrodes. This mixture was compressed under conditions identical to those of the heat-roll press method.

The crystallographic structure of the S-C composite positive electrodes was characterized by XRD using a Cu-Kα radiation source (PANalytical Aeris, Malvern; λ = 1.54060 Å, 40 kV, 15 mA). For the peel test, RT- and 80 °C-pressed S-C composite electrodes were attached to a glass slide using glue (Super Glue, 15187). The uncoated side of the current collector was facing the slide glass. Then, adhesive tape (3 M Scotch® 600) was firmly attached to the top surface of each electrode. A tensile test machine (AEL-A-100, Wenzhou Tripod Instrument Manufacturing Co., Ltd.) with a 50 N load cell was used to conduct the peel test. The tensile tester was used to peel the tape and separate the S-C composite electrode film from the current collector measuring peel force at a speed of 50 μm s$^{-1}$.

The bulk porosity of the electrode is estimated by calculating the ratio between the theoretical densities of sulfur and carbon and the measured density of the fabricated electrodes. To analyze surface voids, the brightness of each image was adjusted to a grayscale range of 62–302, and the images were converted to 8-bit format to analyze surface voids. A manual contrast threshold adjustment was applied to each image to clearly distinguish the voids from filled areas. The percentage of voids present was then calculated by quantifying the surface voids through area analysis. The surface roughness of the material was analyzed using a Surface Profiler (VK-X3000 3D Keyence). This analysis was conducted in optical and laser mode at a magnification of ×50, while scanning an area of 200 × 300 μm. The data obtained from the scan was then processed using the VK-X3000 software to perform a detailed analysis of the surface topology.

The 3D microstructure of the prepared RT- and 80 °C-pressed S-C composite electrodes and slurry-cast S-C composite electrodes were characterized using X-ray μ-CT (Skyscan Bruker). The imaging was conducted at 50 kV and 200 μA with a 4 k resolution and no filter applied. A total of 3600 2D projections were captured over a full 360° rotation of the sample, with an exposure time of 9.25 s for each projection. The 3D-CT images were reconstructed using VGSTUDIO MAX (Volume Graphics) from a series of acquired 2D X-ray projection images. The 3D models were analyzed and visualized using Avizo software (Thermo Fisher). To prepare cross-sectional SEM images of RT- and 80 °C-pressed S-C composite electrodes, the electrodes were vertically embedded in carbon black-infused epoxy (Allied Materials).

After curing overnight, the embedded samples were ground and polished using a polishing wheel. The mechanically polished samples were then further polished using ion milling (Clean Mill Ion Polisher, Thermo Fisher) at 5 kV for 5 h under cryogenic conditions.

To evaluate the wetting properties of the electrode, a precise 7 μL droplet of electrolyte was carefully deposited onto the electrode surface using a micropipette. The entire spreading process was recorded at 1000 frames per second using a high-speed camera to capture the detailed dynamics of the droplet. Subsequently, the contact angle formed by the droplet on the electrode surface was analyzed from the exported images to precisely evaluate and understand the wetting behavior of the electrode.

## Electrochemical test

All electrochemical tests were performed at 25 °C. Each electrochemical condition was tested using multiple independent cells (typically $n = 2$–3, depending on the experiment), and consistent trends were observed; the data shown are representative of typical performance. 2032-type coin cells made of stainless steel (Wellcos) were assembled in an Ar-purged glove box. The cells consisted of lithium foil (120 μm, Honjo Metal) negative electrode, S-C composite positive electrode (single-side coated), a carbon-coated Celgard 2400 (carbon coating: 6 μm and 0.06 mg cm$^{-2}$; Celgard: 25 μm and 1.7 mg cm$^{-2}$), and an electrolyte composed of a 1:1 mixture of Dioxolane (DOL, 99.8%, Sigma-Aldrich) and Dimethoxyethane (DME, 99.5+ %, Sigma-Aldrich), containing 1 M Lithium bis(trifluoromethanesulfonyl) imide (LiTFSI, 99.95 % trace metal basis, Sigma-Aldrich) and 3 wt.% lithium nitrate (LiNO$_3$, 99 + %, Strem). Before use, the salts were dried under vacuum in the glovebox antechamber at 50 °C for 12 h and then immediately transferred into an Ar-filled glovebox. The solvents were dried over molecular sieves (4 Å, Wisesorb) for 48 h prior to use. The electrolyte-to-sulfur ratio (E/S) was set to 10 for 1 mg$_S$ cm$^{-2}$ electrodes, 7 for 3 mg$_S$ cm$^{-2}$, and 5 for 5 mg$_S$ cm$^{-2}$ electrodes. For single-layer pouch cells, the 80 °C-pressed electrode (1.5 × 1.5 cm$^2$, 3 mg$_S$ cm$^{-2}$) was assembled together with the carbon-coated Celgard 2400, Li metal negative electrode (120 μm, Honjo Metal), and the same electrolyte with the E/S ratio of 7 as in the coin cells. Negative to positive (N/P) ratio of the single-layer pouch cell is 4.9. An Al tab (MTI Corporation) was ultrasonically welded onto the as-prepared positive electrode. Li metal was laminated onto a nickel mesh (MTI Corporation) with a pre-welded nickel tab (MTI Corporation). After stacking the electrodes with the carbon-coated separator, the stack was placed in an aluminum-laminated pouch (Wellcos), and the electrolyte was injected using a micropipette in an Ar-filled glovebox to ensure complete wetting of the electrodes and separator. The pouch was then hermetically heat-sealed. No external pressure was applied during cycling. For comparison with the binder-free S-C composite positive electrodes with an areal sulfur loading of 3 mg$_S$ cm$^{-2}$ (70 wt.% sulfur), a S-C composite positive electrode was also prepared using the conventional slurry casting process with a polymer binder. A mixture of S-C composite, Super P (99 + %, Timcal), and LA133 (MSE Supply) binder, at a weight ratio of 83:10:7 (58.1 wt.% active sulfur), was blended using a centrifuge mixer (Thinky ARE-310) with water as the solvent at a solid-to-liquid ratio of 100 mg mL$^{-1}$ for 1 h. The slurry was cast onto carbon-coated Al foil (Wellcos, thickness: 20 μm) with a doctor blade (BYK instrument), air-dried overnight in a fume hood, and subsequently vacuum-dried at 50 °C for 12 h. Lithium metal foil was punched into 15 mm diameter disks inside an Ar-filled glovebox and used immediately as the negative electrode. Galvanostatic charge-discharge cycling and rate performance were tested within a cell voltage window of 1.7–2.7 V using an Arbin LBT Series battery tester, with the cells placed in a multi-zone temperature chamber (Arbin MZTC) maintained at 25 °C. 1 C rate corresponds to 1675 mA g$_S^{-1}$ (based on sulfur mass) in the positive electrode. CV measurements were performed using a potentiostat (BioLogic VMP) within a cell voltage range of 1.7–2.7 V

with scan rates of 0.05 and 0.1 mV s$^{-1}$. EIS was conducted from 1 MHz to 10 mHz using a 5 mV stimulus on before and fully discharged cells after 30 galvanostatic charge-discharge cycles.

S||Li$_2$S symmetric cells were fabricated as follows. Since both electrodes are based on S-C composites and thus lack a native lithium source, the lithiated positive electrode was prepared by fully discharging a Li||S cell to convert sulfur into Li$_2$S prior to assembly. First, Li||S cells were assembled using RT-pressed or 80 °C-pressed electrodes (3 mg$_S$ cm$^{-2}$ areal sulfur loading) and Li metal negative electrodes, with the same electrolyte composition used throughout this study. The cells were discharged at 0.05 °C to 1.7 V to convert sulfur to Li$_2$S, electrochemically, after which the discharged cells were disassembled in an Ar-filled glovebox to recover the lithiated positive electrodes. A symmetric cell was then assembled using the collected Li$_2$S positive electrode as the positive electrode and a fresh S-C positive electrode (identical in mass loading and fabrication condition) as the negative electrode. The symmetric cells were rested for 24 h before galvanostatic cycling at 0.2 °C between −1 V and 1 V, using a 2-h cutoff for both charge and discharge steps.

## In situ OM and Ex situ SEM analysis

In situ OM analysis was performed to examine the dynamic behavior of the positive electrodes. Customized optical cells, each featuring a 3 mm diameter hole, were sealed with a transparent glass window to enable direct observation of the electrode under light irradiation. These optical cells contained side-by-side S-C composite electrodes (3 mg$_S$ cm$^{-2}$) and lithium foil (thickness: 350 μm, Wellcos), filled with a 1 M LiTFSI and 3 wt.% LiNO$_3$ solution in a 1:1 mixture of DOL/DME. Galvanostatic charge-discharge tests were conducted at 0.1 °C within a cell voltage range of 1.8–2.7 V using a WonATech battery tester at 25 °C. All optical cells were assembled in an Ar-filled glove box. In situ OM analysis was carried out with an LV150N optical microscope (Nikon) equipped with a ×50 magnifying lens. Ex situ SEM was performed on electrodes obtained at approximately 40% DOD during the initial discharge at a C/20 (target capacity: 520 mAh g$^{-1}$, based on a discharge capacity of 1300 mAh g$^{-1}$). Cells were disassembled inside an Ar-filled glove box immediately after reaching the targeted discharge capacity. The recovered positive electrodes were gently rinsed with anhydrous DOL/DME mixture solution to remove residual electrolyte and lithium salts, followed by vacuum drying at RT for 2 h. After drying, the electrodes were mounted on SEM stubs inside the Ar-filled glovebox, placed in a sealed glass vial (Parafilm-sealed), and transported to the SEM facility using a double zip-bag to minimize air exposure prior to loading. For cycled electrodes, ex situ SEM samples were collected from cells at the fully charged status after five charge-discharge cycles at 0.2 °C within 1.7–2.7 V. Sample preparation was identical to that described above. Surface morphology was examined using a field-emission scanning electron microscope (Auriga Zeiss) operated at 5 kV accelerating voltage.

## Data availability

All data supporting the findings of this study are available within the article and its Supplementary Information. The numerical data generated in this study that underlie the graphs in the main Figures and Supplementary Figs. are provided in the Source Data file with this paper. Source data are provided with this paper.

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

## Acknowledgements

This research was funded by LG Energy Solution Battery Innovation Contest (Recipient: YH). The authors gratefully acknowledge the use of facilities within the Eyring Materials Center (supported in part by NNCI-ECCS-1542160) and Advanced Electronics and Photonics Core Research Facility at Arizona State University. This research was also partially supported by the National Research Foundation of Korea (NRF) grant funded by the Korea government (MSIT) (RS-2024-00455177, RS-2024-00422387, and RS-2025-00518953, Recipient: S.-H.Y.). We sincerely thank Jordan Monroe and Kaushik Kethamukkala at Arizona State University for their assistance with particle size analysis and X-ray micro-CT, respectively.

## Author contributions

Y.A. and K.K. contributed equally to this work. All authors have given approval to the final version of the manuscript. Y.H. conceptualized the research design. Y.A. and K.K. performed experiments on S-C composite materials and conducted the data curation, formal analysis, investigation, methodology, visualization, and electrochemical measurement of Li‖S batteries. S.K. performed characteristics on SEM and EDS data. F.E. and B.K. conducted thermal compression model simulations. Y.-J.L. and S.-H.Y. designed and performed in situ optical microscope experiment. Y.A. conducted the X-ray CT. Y.A., K.K., and Y.H. wrote the draft. Y.H. performed data curation, funding acquisition, project administration, supervision, writing–original draft, writing–review, and editing.

## Competing interests

The authors declare no competing interests.
