## [Transparent Peer Review file · Nature Communications]

Binding properties of sulfur to enable solvent-free fabrication of high-performance polymer-free sulfur-carbon positive electrodes

Corresponding Author: Professor Yoon Hwa

Version 0:

Reviewer comments:

Reviewer #1

(Remarks to the Author)

The contribution by An et al reports on the preparation of S cathode for LiS batteries based on powder processing without polymer binder. The sulfur is used as binder in this case.

Overall the contribution is timely and well written

The materials are well characterized by various methods including SEM, XRD etc. and electrochemical characterization is extensive.

A major question is the stability after cycling as the binder Sulfur is consumed and should transform to Li_2S . Post mortem analyses could be more comprehensive in this sense to shine light on this aspect.

The crystallographic analysis of sulfur is not so clear. The authors should state which modifications of sulfur are formed in the electrode (alpha, beta etc.). The peak shift is not explained well. What is the origin? This seems to be an artefact. It is recommended to measure with a NIST Si internal standard to confirm the peak shift.

Moreover, the amount of amorphous sulfur should be quantified.

L 214: "Improving transport" is a vague statement

L243 "faster absorptivity" is very qualitative: provide numbers on wetting behavior like contact angle.

It would be good to also analyze electrode swelling after electrolyte wetting vs. dry electrode. KB electrodes significantly expand typically with DME/DOL.

L287: cycling behavior. This probably reflects more the instability of Li anode than cathode. A better way to monitor cathode cycling stability alone is to work with symmetric cathode cells.

L303: RT-pressed cells are a poor benchmark. Its easy to outperform a bad standard. Refer to literature and a standard electrode with PTFE binder to compare in fair way.

Fig 4: low loading in cathode makes it easy to get high cycling stability. Only 1 mg/cm² and E/S = 10!

This will never give a high specific energy. E/S < 3 needed and at least 3 mg/cm². Please also provide data for higher loading and lower E/S. Pouch cell data would be better. Limitations of coin cells are well documented in literature but not cited.

Methods: wavelength of Ag radiation should be given and also mentioned in XRD figure for clarification.

References:

Critical references (reviews) on Dry processing as well as LiS batteries are missing! Literature too narrow.

Reviewer #2

(Remarks to the Author)

Summary: This Research Paper NCOMMS-25-41503, entitled "Unveiling sulfur as a binder for high-performance polymer-free sulfur-carbon cathodes via solvent-free fabrication," reports a novel, scalable, and environmentally friendly method for fabricating binder- and solvent-free sulfur-carbon composite cathodes for lithium-sulfur cells via thermal-assisted dry pressing directly onto aluminum foil. By using sulfur's intrinsic softening and adhesive properties, the process eliminates the

need for polymeric binders and toxic solvents. Thus, the heat-roll pressed cathode simplifies the electrode fabrication and achieves a high reversible capacity of 932 mAh g⁻¹ over 500 cycles at a 1C rate. The use of low-cost commercial porous carbon further enhances the practical relevance of the approach, which also reduces production costs by more than 50%.

[General Comment]: This research reports the novel fabrication design and optimization for lithium–sulfur battery cathodes with comprehensive analysis and study. The following minor revisions are suggested to further improve the clarity and impact of the work.

Additional Comments:

(1) For the fundamental concept, sulfur would be served as both the active material and the binder in the heat-roll pressed cathode. During cycling sulfur becomes liquid-state polysulfides, this solid-liquid conversion would weaken the electrode structure. Moreover, polysulfides would convert to sulfur and sulfide during charge and discharge. The relocation and reformation of solid-state active material during cycling do not have heat and pressure to support the stability of the heat-roll pressed cathode. These concerns can be discussed.

[Suggestion] Please consider having a discussion on the cathode stability during polysulfide formation and its nucleation toward sulfur and sulfide.

(2) In the electrochemical analysis, the discharge/charge efficiency of the heat-roll pressed cathode would decrease during cycling and would decrease to less than 90%. However, 5 mg cm⁻² cathode shows the discharge/charge efficiency of over 100%. These issues are suggested to be addressed and discussed. With this concern, discharge/charge efficiency is suggested to be reported in all electrochemical analysis, including cycling and rate performances.

[Suggestion] Please address the issues of discharge/charge efficiency, including the missing data, the decrease in electrochemical efficiency, and the efficiency of over 100%.

(3) In the electrochemical analysis, the electrochemical impedance analysis of the heat-roll pressed cathode is suggested to be analyzed with an equivalent circuit model to support the corresponding discussion. The raw experimental data and the fitting data are suggested to be reported together as experimental data points and the fitting curves.

[Suggestion] Please analyze the impedance data.

(4) The heat-roll pressed cathode is reported with the sulfur loadings of 1 to 5 mg cm⁻². The cell-fabrication parameters of 5 mg cm⁻² cathode cannot be found in the manuscript. The corresponding sulfur loading and content and the related electrolyte-to-sulfur ratio are suggested to be reported together for a reference and for a comparison. The amount of lithium is suggested to be reported.

[Suggestion] Please report on the necessary cell-fabrication parameters.

(5) After reviewing the introduction and references, it is suggested to provide a more comprehensive literature survey. Several recent studies have reported cathode fabrication methods involving heat and pressing treatments, many of which are closely related to the present work. Incorporating a discussion of these studies could help position this research within the current landscape and highlight its novelty.

[Suggestion] Please consider expanding the introduction and discussion sections to include recent advances in cathode design using thermal and pressure-assisted treatments.

Reviewer #3

(Remarks to the Author)

This manuscript presents a solvent- and binder-free approach for fabricating sulfur-carbon composite cathodes via thermally assisted dry-pressing, leveraging sulfur's softening behavior to serve as an intrinsic binder. The experimental work is well executed and supported by structural characterizations, in-situ observations, and mechanical simulations. The study addresses a practical and timely issue in green and scalable manufacturing of Li–S batteries, achieving long cycle life and competitive capacity metrics. However, the conceptual novelty of the work is limited. The central idea is largely based on the well-known thermal softening behavior of sulfur and does not introduce fundamentally new mechanisms in materials design or electrochemistry. Moreover, the role of sulfur as a persistent structural binder under cycling conditions is not convincingly demonstrated. The manuscript also lacks a clear differentiation from existing binder-free cathode strategies. In its current form, the work does not meet the originality and mechanistic depth expected for publication in Nature Communications.

Comments

Comment 1. The manuscript lacks experimental or theoretical validation that sulfur retains binding capability after being converted into soluble polysulfides.

Comment 2. While ex-situ images demonstrate morphological integrity, they do not adequately explain how the electrode resists internal fracture or delamination under repeated volume changes.

Comment 3. Comparison with prior binder-free strategies is insufficient. Previous studies using melt-infiltration, porous carbons, or 3D scaffolds have pursued similar goals.

Comment 4. There is no quantitative modeling or correlation to electrochemical performance metrics such as impedance, diffusion kinetics, or polarization behavior. This weakens the causal link between electrode structure and performance.

Comment 5. The manuscript emphasizes the industrial potential of the method but omits key details such as roll-to-roll process compatibility, throughput, or large-area uniformity. More discussion or preliminary validation would be necessary.

Comment 6. The SEM image in Figure 2 is difficult to interpret. It is recommended to include both a magnified view and a lower-magnification overview to aid in visual comprehension of the particle morphology and packing behavior.

Reviewer #4

(Remarks to the Author)

This work presents a new way to fabricate solvent-free Li/S batteries with reduced fabrication costs and improved performance. The work is interesting and well written. However, to be accepted by Nature Communications, the following issues should be considered.

1. Figure 1 b, clarify the procedure you used to characterise the powder rheology – usually this is performed by shear cell, or FT4.
2. Line 125, clarify if you are using 50 mm powder bed - This seems strange to describe the thin electrode behaviour.
3. Figure 1 g-j,
 - (1) Are you using ROCKY to quantify the particle deformation? please clarify if FEM approach is used in ROCKY module, if yes, can you extract the pressure acting on the punch? to calibrate your simulation results.
 - (2) need to specify the simulation parameters and calibration procedure of bond model used for the simulation.
 - (3) the spherical particle approximation may not be suitable to describe sulphur particles.
 - (4) Line 407-408, where did you find the general relationships of particle property and temperature.
In general, this part needs more experimental/theoretical support to make sure your finding sound enough!
4. Figure 4, will the compressing level significantly affect the battery performance?
5. By using the CT-scan microstructures, do the authors need to add some transport/electrochemical simulation results to indicate the mesoscale effect on the electrochemical property?

Minor errors

1. Table S1, 'kg' should be deleted for the amount of elemental sulfur.

Version 1:

Reviewer comments:

Reviewer #1

(Remarks to the Author)

The authors have added new data. However in some cases, only literature was added. Symmetrical cathode tests are added which is valuable.

Amorphous content: PDF only gives limited insight. Normally, amorphous content is determined with internal intensity standard.

Benchmark: The authors should measure a benchmark electrode with liquid electrolyte for comparison themselves. A solid state battery is not a proper benchmark.

The authors should provide pouch cell data, not only refer to literature data.

Reviewer #2

(Remarks to the Author)

The paper's quality has improved significantly since its revision. By incorporating all the suggestions, the authors produced a revised manuscript that is now recommended for publication.

Reviewer #3

(Remarks to the Author)

The authors have carefully addressed the earlier concerns by adding new experimental data (ex-situ SEM, EIS fitting, high-magnification SEM) and expanding the discussion on structural stability, binder-free comparisons, and scalability. While the conceptual novelty is incremental, the study provides a practical and well-supported approach toward solvent- and binder-free Li-S cathodes, with convincing experimental validation.

Reviewer #4

(Remarks to the Author)

The revisions are substantial, however the following two questions needs to be answered:

1. Clarify if you are using 5 μm or 5 mm particles in the powder bed simulation, from powder technology viewpoint, because of adhesive force difference, the bulk powder behavior could be totally different.
2. Are you using the same bond parameters in the bond model to model the powder compaction behaviour of RT and 80 °C? If this is the case, can you justify what material you need to describe use bond model in your compaction simulation? is it

reasonable to use same baon parameters under same conditions?

Version 2:

Reviewer comments:

Reviewer #1

(Remarks to the Author)

The referee commenst are carefully addressed.

Before publication, the E/S ratio in the pouch cell should be given. Actually, the experimental section on the pouch cell is quite limited and could be extended in the ESI.

Reviewer #4

(Remarks to the Author)

This work is sututable to be accepted now.

Responses to Reviewers' Comments

Responses to the reviewer's comments:

In the following pages, we have provided detailed and complete responses to the reviewer's comments and observations made to the manuscript (Research Article, No. NCOMMS-25-41503) entitled, "Unveiling Sulfur as a Binder for High-Performance Polymer-Free Sulfur-Carbon Cathodes via Solvent-Free Fabrication". The revised or newly added contents have been highlighted in yellow in both the revised Manuscript and Supplementary Information. These changes are also marked in yellow in this response to the reviewers' comments.

Reviewer #1

Overall comment: The contribution by An et al reports on the preparation of S cathode for Li/S batteries based on powder processing without polymer binder. The sulfur is used as binder in this case. Overall the contribution is timely and well written. The materials are well characterized by various methods including SEM, XRD etc. and electrochemical characterization is extensive. A major question is the stability after cycling as the binder Sulfur is consumed and should transform to Li_2S . Post mortem analyses could be more comprehensive in this sense to shine light on this aspect.

Response: We greatly appreciate the reviewer for positive comments. We hope that our revisions have addressed the reviewer's concerns. Point-to-point replies to the reviewer's comments are listed below.

Comment 1-1: The crystallographic analysis of sulfur is not so clear. The authors should state which modifications of sulfur are formed in the electrode (alpha, beta etc.). The peak shift is not explained well. What is the origin? This seems to be an artefact. It is recommended to measure with a NIST Si internal standard to confirm the peak shift. Moreover, the amount of amorphous sulfur should be quantified.

Response: We appreciate the reviewer's feedback regarding the crystallographic analysis of sulfur. In our study, the diffraction peaks observed in all samples match well with the JCPDS card No. 08-0247, which corresponds to orthorhombic α -sulfur (S_8)—the thermodynamically stable allotrope at ambient conditions. To address concerns about peak alignment, we used the diffraction peak of the carbon-coated Al substrate (marked with green diamonds in Fig. 3a) as an internal reference to align the patterns, rather than a standard Si, given the binder-free electrode configuration. The observed slight shift of the (222) peak upon pressing at 80 °C compared to that of RT-pressed cathodes is attributed to thermomechanical strain or local

distortion within the α -sulfur lattice during compression. To clarify this point, the following statements have been revised or newly added in the revised manuscript.

- In Result and Discussion section of the revised Manuscript:
XRD (Fig. S6) shows crystalline α -sulfur (JCPDS 08-0247) in the composite,
- In Result and Discussion section of the revised Manuscript:
The shift and associated peak shape distortion of α -sulfur in 80 °C-pressed cathodes suggest thermomechanical strain or local lattice distortion within α -sulfur, likely facilitating improved particle–particle cohesion and mechanical integrity in the composite electrode.

To further investigate the structural state of sulfur in the S–C composite, we performed laboratory-based total scattering measurements and pair distribution function (PDF) analysis (Fig. S7). While direct quantification is still challenging due to the confinement of sulfur within the porous carbon matrix, we made efforts to investigate the presence of non-crystalline sulfur. The broadened features in $I(Q)$ and $S(Q)$, along with the attenuation of long-range correlations in $G(r)$, clearly indicate partial amorphization of sulfur within the composite. Although this method does not allow precise quantification of amorphous content, it supports the existence of amorphous sulfur domains confined within the porous host. The following statements and references have been incorporated into the revised manuscript, and the corresponding PDF results and discussion have been included in the revised Supplementary Information.

- In Results and Discussion section of the revised Manuscript:
and to further investigate local structural features including non-crystalline sulfur, we performed pair distribution function (PDF) analysis for elemental sulfur and S-C composite (Fig. S7). Compared to elemental sulfur, the S–C composite exhibited broadened features in $I(Q)$ and $S(Q)$, [36,37] along with diminished long-range correlations in $G(r)$, [38,39] indicating the presence of amorphous sulfur domains. The preservation of nearest-neighbor S–S distances further suggests that local sulfur bonding is retained despite structural disorder induced by confinement within the porous carbon matrix.
- In References section of the revised Manuscript:
36. Zhang, L. *et al.* Chanin breakage in the supercooled liquid-liquid transition and re-entry of the λ -transition in sulfur. *Sci. Rep.* **8**, 4558 (2018).
37. Winter, R., Egelstaff, P. A., Pilgrim, W.-C. & Howells, W. The structural properties of liquid, solid and amorphous sulphur. *J. Phys.: Condens. Matter* **2**, SA215–SA218 (1990).
38. Shiotani, S. *et al.* Pair distribution function analysis of sulfide glassy electrolytes for all-solid-state batteries: Understanding the improvement of ionic conductivity under annealing condition. *Sci. Rep.* **7**, 6972 (2017).
[39] Yamaguchi, H. *et al.* Local structure of amorphous sulfur in carbon-sulfur composites for all-solid-state lithium-sulfur batteries. *Commun. Chem.* **8**, 10 (2025).
- In Methods section of the revised manuscript:
PDF measurements were performed using a STOE STADI 60 P transmission X-ray diffractometer equipped with an Ag $K\alpha_1$ source ($\lambda = 0.55941 \text{ \AA}$; 40 kV, 40 mA) and a

curved Ge(111) monochromator. Diffraction patterns were collected in Debye–Scherrer geometry using four stationary MYTHEN2 R 1K silicon strip detectors covering a 2θ range of 2° – 140° ($Q_{\max} = 21 \text{ \AA}^{-1}$). Samples were sealed in spinning borosilicate glass capillaries (0.50 mm outer diameter, 10 μm wall thickness) to minimize preferred orientation and enhance statistical averaging. Each sample was measured for 24 h, and background scattering was subtracted using data from an identically measured empty capillary. The 1D diffraction patterns were processed with PDFgetX3 to yield the background-subtracted intensity $I(Q)$, structure function $S(Q)$, reduced function $F(Q)$, and real-space pair distribution function $G(r)$. [39,62]

- In References section of the revised Manuscript:

39. Yamaguchi, H. *et al.* Local structure of amorphous sulfur in carbon-sulfur composites for all-solid-state lithium-sulfur batteries. *Commun. Chem.* **8**, 10 (2025).

62. Juhás, P., Davis, T., Farrow, C. L. & Billinge, S. J. L. PDFgetX3: a rapid and highly automatable program for processing powder diffraction data into total scattering pair distribution functions. *J. Appl. Cryst.* **46**, 560–566 (2013)

- In the revised Supplementary Information:

Figure S7 Pair distribution function analysis comparing elemental sulfur (black) and the synthesized S–C composite (red): a) background-subtracted scattering intensity $I(Q)$, b) total scattering structure function $S(Q)$, c) reduced structure function $F(Q)$, d) real-space pair distribution function $G(r)$.

- In the revised Supplementary Information:

The diffraction profiles of the melt-diffused S/C composite exhibit broadened features in $I(Q)$ and $S(Q)$ relative to the sharp Bragg peaks observed in elemental sulfur, particularly around $q \approx 1.7 \text{ \AA}^{-1}$. This broadening signifies partial amorphization of sulfur as it diffuses into the porous carbon framework. Further evidence is provided by the real-space pair distribution function, $G(r)$, where the melt-diffused sample displays a notable attenuation of long-range correlations beyond $\sim 5 \text{ \AA}$, in contrast to the extended order seen in elemental

sulfur. Nevertheless, the characteristic nearest-neighbor S–S distances at 2.1 Å, 3.4 Å, and 4.5 Å are preserved, indicating the local S–S bonding is maintained. Additionally, a minor distance at 1.4 Å appears exclusively in the melt-diffused sample, which is attributed to the carbon framework in the S-C composite. These results collectively confirm the presence of amorphous sulfur domains formed through melt infiltration and confinement within the porous carbon matrix.

Comment 1-2: “Improving transport” is a vague statement.

Response: We appreciate the reviewer’s comment regarding the need to clarify the phrase “improving transport.” We have revised the text to specify that the reduction in tortuosity observed in the 80 °C-pressed cathode is expected to facilitate Li-ion migration by providing more direct ion pathways and minimizing diffusion resistance. The following statement have been added to the revised manuscript.

- In Results and Discussion section of the revised Manuscript:

The reduction in tortuosity is expected to facilitate more efficient Li-ion migration by shortening diffusion paths, which can contribute to improved rate capability and electrochemical uniformity during cycling.

Comment 1-3: “faster absorptivity” is very qualitative: provide numbers on wetting behavior like contact angle. It would be good to also analyze electrode swelling after electrolyte wetting vs. dry electrode. KB electrodes significantly expand typically with DME/DOL.

Response: Thank you for the valuable comments. We appreciate the reviewer’s careful reading and constructive suggestions, which helped improve the clarity and scientific rigor of the manuscript. In response to the comment, we have revised the relevant sentence to provide additional detail and quantitative context. The revised statement has been incorporated into the manuscript, and the wetting test images after 2 s have been updated in the revised Supplementary Information.

- In Results and Discussion section of the revised Manuscript:

The binder-free cathodes display more rapid and uniform electrolyte absorption, with contact angles of 17° and 14° after 0.02 s for RT-pressed and 80 °C-pressed samples, respectively, compared to 23° for the slurry-cast S–C composite cathode. This enhanced wettability is attributed to their more homogeneous porosity and structural integrity (Fig. S19).

- In Result and Discussion section of the revised Manuscript:

After 2 s, the RT-pressed electrode shows surface swelling and irregularity (Fig. S19), indicating localized expansion and reduced mechanical stability. In contrast, the 80 °C-pressed and slurry-cast electrodes maintain flat, intact surfaces, suggesting better structural resilience.

- In the revised Supplementary Information:

Figure S19. Electrolyte wetting experiment on the a) RT-pressed, b) 80 °C-pressed, and c) slurry-cast S-C composite cathodes using a high-speed camera.

Comment 1-4: Cycling behavior. This probably reflects more the instability of Li anode than cathode. A better way to monitor cathode cycling stability alone is to work with symmetric cathode cells.

Response: We agree with the reviewer that conventional Li/S cell cycling can obscure cathode-specific behavior due to concurrent instability of the Li metal anode. To address this, we performed galvanostatic cycling of S/Li₂S symmetric cells, in which a lithiated cathode (Li₂S-containing) was paired with a fresh S/C cathode under identical electrolyte and cell configuration conditions (Fig. S24). Since both electrodes are based on S/C composites and thus lack a native lithium source, the lithiated cathode was prepared by fully discharging a Li/S cell to convert sulfur into Li₂S prior to assembly. This configuration isolates cathode behavior by eliminating the influence of a Li-metal anode. To address this, we performed galvanostatic cycling of S/Li₂S cells in which a lithiated cathode (Li₂S-containing) was paired with a fresh S/C cathode under identical electrolyte and cell configuration conditions (**Fig. S24**). This configuration isolates cathode-side phenomena by removing contributions from Li metal. The symmetric cell results reveal a clear distinction between RT-pressed and 80 °C-pressed electrodes. The RT-pressed electrode shows high initial charge overpotentials and premature discharge cutoff, indicating sluggish Li₂S oxidation and limited reversibility. In contrast, the 80 °C-pressed electrode exhibits low charge polarization from the first cycle and progressive

improvement in discharge behavior, suggesting enhanced electrochemical reversibility. These results are consistent with the superior structural cohesion, improved interparticle connectivity, and more uniform Li₂S distribution previously observed in the 80 °C-pressed samples. The following statements and figures have been added in the revised manuscript and supplementary information, respectively.

- In Results and Discussion section of the revised Manuscript:

S/Li₂S cell testing (Fig. S24) in which a lithiated cathode (Li₂S-containing) was paired with a fresh S/C cathode further confirmed enhanced reversibility in 80 °C-pressed cathodes, which exhibited lower charge polarization and more stable cycling behavior compared to RT-pressed samples, consistent with improved structural integrity and Li₂S distribution.

- In Methods section of the revised Manuscript:

S/Li₂S symmetric cells were fabricated as follows. Since both electrodes are based on S/C composites and thus lack a native lithium source, the lithiated cathode was prepared by fully discharging a Li/S cell to convert sulfur into Li₂S prior to assembly. First, Li/S cells were assembled using RT-pressed or 80 °C-pressed cathodes (3 mg cm⁻² areal mass loading) and Li metal anodes, with the same electrolyte composition used throughout this study. The cells were discharged to 1.7 V (vs. Li⁺/Li) to convert sulfur to Li₂S, electrochemically, after which the discharged cells were disassembled in an Ar-filled glovebox to recover the lithiated cathodes. A new symmetric cell was then assembled using the collected Li₂S cathode as the positive electrode and a fresh S/C cathode (identical in mass loading and fabrication condition) as the negative electrode. The symmetric cells were rested for 24 h before galvanostatic cycling at 0.2 C (1 C = 1675 mAh g⁻¹) between -1 V and 1 V, using a 2-hour cutoff for both charge and discharge steps.

Figure S24. Galvanostatic cycling profiles of S-Li₂S symmetric cells in which a lithiated cathode (Li₂S-containing) was paired with a fresh S/C cathode. Tests were conducted at 0.2 C (1 C = 1675 mAh g⁻¹) within -1 V to 1 V, with a 2 h cutoff per step.

Comment 1-5: RT-pressed cells are a poor benchmark. It's easy to outperform a bad standard. Refer to literature and a standard electrode with PTFE binder to compare in fair way.

Response: We appreciate the reviewer's comment regarding the use of RT-pressed cathodes as a benchmark. Our intent in including the RT-pressed cathode was not to serve as a low baseline

for performance comparison, but rather to evaluate the impact of processing temperature—an important parameter in solvent-free electrode fabrication—on electrode structure and electrochemical properties. The contrast between RT- and 80 °C-pressed electrodes allows us to systematically investigate how thermomechanical effects during compression influence porosity, tortuosity, interfacial contact, and ultimately cell performance.

To ensure fair and meaningful comparison with state-of-the-art sulfur cathodes, we previously compiled a summary of recent literature in Table S2 in the original manuscript, which includes key electrochemical performance metrics (initial capacity, cycling stability, sulfur loading, binder type, etc.). Our composite cathode shows competitive or superior cycling stability across a range of sulfur loadings (1, 3, and 5 mg cm⁻²) and E/S ratios, even without using polymeric binders. As the reviewer correctly pointed out, however, PTFE-based electrodes—which are particularly relevant for dry-processing—were previously missing from this comparison table. We have now added representative PTFE-based sulfur cathode studies to Table S2 to more comprehensively contextualize our results. These additions help to further validate the strong performance of our binder-free, solvent-free processed electrodes relative to both slurry-cast and dry-processed counterparts. The following articles have been cited in Table S2 in the revised Supplementary Information.

- In Supporting Reference section of the revised Supplementary Information:

[S15] Hu, J.-K. *et al.* Dry electrode technology for scalable and flexible high-energy sulfur cathodes in all-solid -state lithium-sulfur batteries. *J. Energy Chem.* **71**, 612–618 (2022).

[S16] Fiedler, M. *et al.* Mechanistic insights into the cycling behavior of sulfur dry-film cathodes. *Adv. Sustainable Syst.* **7**, 2200439 (2023).

[S17] Sul, H., Lee, D. & Manthiram, A. High-loading lithium-sulfur batteries with solvent-free dry-electrode processing. *Small* **20**, 2400728 (2024).

Comment 1-6: low loading in cathode makes it easy to get high cycling stability. Only 1 mg/cm² and E/S = 10! This will never give a high specific energy. E/S < 3 needed and at least 3 mg/cm². Please also provide data for higher loading and lower E/S. Pouch cell data would be better. Limitations of coin cells are well documented in literature but not cited.

Response: We thank the reviewer for highlighting the limitations of low-loading electrodes and high electrolyte-to-sulfur (E/S) ratios in Li/S cells. We fully agree that achieving practically relevant specific energy requires high sulfur loading (>3 mgs cm⁻²) and low E/S ratios (<3 μL mg⁻¹). In this study, in addition to 1 and 3 mgs cm⁻² cathodes with E/S ratio of 7 and 5, respectively, (Figure 4) we also evaluated higher-loading electrodes (5 mgs cm⁻²) with E/S ratios of 5, as presented and Fig. S21 of the revised manuscript (Figures are shown below). The 80 °C-pressed cathodes with those areal loadings demonstrated cycling stability over 200 cycles and 50 cycles for the 3 mg cm⁻² and 5 mg cm⁻² electrodes, respectively. While the E/S ratios are not as low as those referenced in the reviewer’s comment, we believe the presented

data support the feasibility and stability of our composite electrode design under more practically relevant conditions. Low-loading cathodes were employed during early-stage evaluation to enable more uniform sulfur distribution, improved electrolyte wetting, and enhanced reaction kinetics, which facilitate the assessment of intrinsic material properties of cathodes.

Figure S21. Electrochemical performance of the slurry cast S-C composite cathodes with areal mass loading of $5 \text{ mg}_S \text{ cm}^{-2}$. a) Rate capability test results, b) cycle performance comparison, voltage profiles of c) RT-pressed, d) 80°C -pressed, and e) slurry-cast S-C composite cathodes at various C-rate ($1 \text{ C} = 1675 \text{ mAh g}^{-1}$) and Nyquist plot of f) RT-pressed, g) 80°C -pressed, and h) slurry-cast S-C composite cathodes, **measured at the fully charged state** before and after 50 charge-discharge cycling.

- In Methods section of the revised Manuscript:

The electrolyte-to-sulfur ratio (E/S) was set to 10 for $1 \text{ mg}_S \text{ cm}^{-2}$ electrodes, 7 for $3 \text{ mg}_S \text{ cm}^{-2}$ and 5 for $5 \text{ mg}_S \text{ cm}^{-2}$ electrodes.

Response: Regarding the use of coin cells, we agree that pouch cells provide more realistic system-level information, especially for evaluating E/S ratios, mechanical constraints, and practical energy density. This has been addressed in the revised manuscript, where we now cite prior literature that discusses the limitations of coin cells in practical relevance studies. The following sentences have been added, and research articles have been cited in the revised manuscript.

- In Results and Discussion section of the revised Manuscript:
While pouch cells are important for assessing practical energy metrics,[40–42] coin cells were used to enable controlled evaluation of the intrinsic properties of the newly developed sulfur cathodes in this work.
- In References section of the revised Manuscript:
40. Xing, C., Chen, H. & Zhang, S. Powering 10-Ah-level Li-S pouch cell via a smart “skin”. *Matter* **5**, 2523–2525 (2022).
41. Das, S. *et al.* Optimization of the form factors of advanced Li-S pouch cells. *Small* **20**, 2311850 (2024).
42. Yari, S., Reis, A. C., Pang, Q. & Safari, M. Performance benchmarking and analysis of lithium-sulfur batteries for next-generation cell design. *Nat. Commun.* **16**, 5473 (2025).

Comment 1-7: Wavelength of Ag radiation should be given and also mentioned in XRD figure for clarification.

Response: We appreciate the reviewer’s comment regarding the missing information. The wavelength of XRD sources have been added in the revised manuscript.

- In Methods section of the revised Manuscript:
Operando XRD (STADI P STOE Dual Transmission, Ag-K α source, $\lambda = 0.55941 \text{ \AA}$, 40 kV, 40 mA) for heating was conducted to study the thermal behavior of elemental sulfur (sulfur nano-powder, Skyspring).
- In Results and Discussion section of the revised Manuscript:
The crystal structure of the S-C composite was scanned between 10 and 70 2 θ degree using Cu-K α source XRD (PANalytical Aeris Malvern, $\lambda = 1.54060 \text{ \AA}$, 40kV, 15mA).
- In Results and Discussion section of the revised Manuscript:
The crystallographic structure of the S–C composite cathodes were characterized by XRD using a Cu K α radiation source (PANalytical Aeris, Malvern; $\lambda = 1.54060 \text{ \AA}$, 40kV, 15mA).

Comment 1-8: Critical references (reviews) on Dry processing as well as Li/S batteries are missing! Literature is too narrow.

Response: We thank the reviewer for pointing out the limited scope of the literature citations related to dry processing and Li/S battery development. In response, we have revised the Manuscript and Supplementary Information to include additional review articles that summarize recent progress and ongoing challenges in both areas. The following sentences have been added and research articles on dry processing of sulfur cathode have been cited in the revised manuscript.

- In Results and Discussion section of the revised Manuscript:

Recent reviews have highlighted growing interest in dry-fabrication of sulfur cathodes for Li/S batteries to improve manufacturing efficiency and interface control.[21–24]

- In References section of the revised Manuscript:

21. Kim, N.-Y. *et al.* Material challenges facing scalable dry-processable battery electrodes. *ACS Energy Lett.* **9**, 5688–5703 (2024).

22. Jin, W. *et al.* Advancements in dry electrode technologies: Towards sustainable and efficient battery manufacturing. *ChemElectroChem* **11**, e202400288 (2024).

23. Hong, T. H., Kim, D. J., Ko, S. M. & Lee, J. T. Solvent-free dry-process for developing high-performance lithium-sulfur batteries. *Korean J. Chem. Eng.* **42**, 1475–1490 (2025).

24. Park, J. *et al.* Sustainable and cost-effective electrode manufacturing for advanced lithium batteries: the roll-to-roll dry coating process. *Chem. Sci.* **16**, 6598–6619 (2025).

Thank you very much for the valuable comments. Your comments significantly improved the quality of the manuscript.

Reviewer #2

Overall comment: This Research Paper NCOMMS-25-41503, entitled “Unveiling sulfur as a binder for high-performance polymer-free sulfur-carbon cathodes via solvent-free fabrication,” reports a novel, scalable, and environmentally friendly method for fabricating binder- and solvent-free sulfur-carbon composite cathodes for lithium-sulfur cells via thermal-assisted dry pressing directly onto aluminum foil. By using sulfur’s intrinsic softening and adhesive properties, the process eliminates the need for polymeric binders and toxic solvents. Thus, the heat-roll pressed cathode simplifies the electrode fabrication and achieves a high reversible capacity of 932 mAh g⁻¹ over 500 cycles at a 1C rate. The use of low-cost commercial porous carbon further enhances the practical relevance of the approach, which also reduces production costs by more than 50%. This research reports the novel fabrication design and optimization for lithium-sulfur battery cathodes with comprehensive analysis and study. The following minor revisions are suggested to further improve the clarity and impact of the work.

Response: We sincerely thank the reviewer for the constructive and encouraging feedback. We have carefully revised the manuscript in response to the comments, and we hope that our revisions satisfactorily address all concerns. Point-by-point responses to each comment are provided below.

Comment 2-1: For the fundamental concept, sulfur would be served as both the active material and the binder in the heat-roll pressed cathode. During cycling sulfur becomes liquid-state polysulfides, this solid-liquid conversion would weaken the electrode structure. Moreover, polysulfides would convert to sulfur and sulfide during charge and discharge. The relocation and reformation of solid-state active material during cycling do not have heat and pressure to support

the stability of the heat-roll pressed cathode. These concerns can be discussed. [Suggestion] Please consider having a discussion on the cathode stability during polysulfide formation and its nucleation toward sulfur and sulfide.

Response: We thank the reviewer for raising an important point regarding the structural stability of the cathode during the solid–liquid–solid transitions associated with polysulfide formation and sulfur re-nucleation. We fully agree that these conversion reactions pose a significant challenge for maintaining electrode stability, especially in binder- and solvent-free configurations. To address this, we have revised the manuscript to include a discussion on how thermomechanical compression at 80 °C enhances the structural robustness of the cathode and mitigates degradation during cycling. In response, we have added a discussion to the revised manuscript highlighting how thermomechanical compression (via 80 °C pressing) contributes to maintaining electrode integrity throughout cycling.

In addition, to further examine whether sulfur retains its mechanical binding functionality even after partial conversion to soluble species, we conducted ex-situ SEM imaging at 40% depth of discharge (DOD). The following sentence has been added in the revised manuscript. These data indicate that the 80 °C-pressed electrodes maintain structural cohesion even at the early stages of electrochemical cycling, while RT-pressed electrodes exhibit surface irregularity and local fracture. As highlighted in literature such as Wujcik et al. [J. Electrochem. Soc. 164, A18-A27 (2017)], sulfur conversion reactions do not proceed uniformly across the entire electrode. Heterogeneous distribution of Li_2S and Li_2S_2 species, along with residual elemental sulfur and soluble polysulfides, may coexist during intermediate cycling stages. This spatial heterogeneity could allow certain domains of the electrode to retain solid-state binding capability, thereby preserving local structural integrity even in the absence of conventional binders. We emphasize that while the current experimental results support the hypothesis that thermomechanically compressed sulfur can retain binding functionality during partial conversion, a complete mechanistic understanding of the solid–liquid–solid transitions and their spatial distribution within the electrode remains an open question. Further investigation—potentially involving in situ or operando characterization techniques combined with mesoscale modeling—will be required to fully elucidate the mechanisms by which structural cohesion is preserved during cycling. The following statement and literature have been added in the revised Manuscript and the ex-situ SEM images of S-C cathodes at 40 % DOD have been added in the revised Supplementary Information.

- In Results and Discussion section of the revised Manuscript:

Additional ex-situ SEM images collected at ~40% depth of discharge at the initial discharge process reveal that the 80 °C-pressed cathodes retain structural cohesion during early electrochemical cycling (Fig. S23). This observation, together with prior findings that sulfur conversion does not occur uniformly across the electrode,[52] suggests that partially unreacted sulfur and early-stage Li_2S_x species can coexist, allowing localized structural retention during the formation of discharge products. Importantly, the observations suggest

that the initial mechanical integrity provided by heat-pressing plays a critical role in mitigating the potential structural weakening associated with the electrochemical solid–liquid–solid transition of sulfur during cycling. Although polysulfide formation and subsequent sulfur precipitation inherently involve relocation of active material, the 80 °C-pressed cathode preserves a cohesive framework that accommodates these transformations without particle detachment or severe cracking, as confirmed by ex-situ SEM. S/Li₂S cell testing (Fig. S24) in which a lithiated cathode (Li₂S-containing) was paired with a fresh S/C cathode further confirmed enhanced reversibility in 80 °C-pressed cathodes, which exhibited lower charge polarization and more stable cycling behavior compared to RT-pressed samples, consistent with improved structural integrity and Li₂S distribution. This indicates that effective thermomechanical compression, by promoting structural uniformity, not only ensures intimate interparticle contact at the outset but also helps preserve the electrode architecture throughout repeated Li/S redox cycles, as summarized in Fig. 5e and 5f.

- In Methods section of the revised Manuscript:

Ex-situ SEM was performed on electrodes obtained at approximately 40% DOD during the initial discharge cycle or after five charge-discharge cycles. For 40 % DOD cathode, coin-type half-cells were disassembled inside an Ar-filled glove box after reaching the targeted discharge capacity. The cathodes were gently rinsed with anhydrous DOL/DME mixture solution to remove residual electrolyte and lithium salts, followed by vacuum drying at RT for 2 hours. Surface morphology was examined using a field-emission scanning electron microscope (Auriga Zeiss) operated at 5 kV accelerating voltage.

- In Reference section of the revised Manuscript:

52. Wujcik, K. H. et al. In situ X-ray absorption spectroscopy studies of discharge reactions in a thick cathode of a lithium sulfur battery. *J. Electrochem. Soc.* 164, A18–A27 (2017).

- In the revised Supplementary Information:

Figure S23. Ex-situ SEM images of S-C cathodes collected at 40% depth of discharge at the initial discharge process. (a) RT-pressed cathode, (b) 80 °C-pressed cathode.

Comment 2-2: In the electrochemical analysis, the discharge/charge efficiency of the heat-roll pressed cathode would decrease during cycling and would decrease to less than 90%. However, 5 mg cm⁻² cathode shows the discharge/charge efficiency of over 100%. These issues are suggested to be addressed and discussed. With this concern, discharge/charge efficiency is suggested to be reported in all electrochemical analysis, including cycling and rate performances. [Suggestion] Please address the issues of discharge/charge efficiency, including the missing data, the decrease in electrochemical efficiency, and the efficiency of over 100%.

Response: We thank the reviewer for the insightful comment regarding the discharge/charge efficiency of the cathodes and the suggestion to report it consistently across all electrochemical analyses. As the reviewer mentioned, the Coulombic efficiency values exceeding 100%, mainly observed in the 80 °C-pressed cathodes with high sulfur loading (5 mg cm⁻²), are not found in the RT-pressed or slurry-cast counterparts. This phenomenon could be attributed to the distinct structural and electrochemical characteristics of the 80 °C-pressed electrodes. Their compact and well-connected microstructure enables more complete sulfur conversion, resulting in higher discharge capacities. In contrast, the RT-pressed and slurry-cast electrodes exhibit poor particle interconnectivity and structural heterogeneity, which limit sulfur utilization. Moreover, the uniform architecture of the 80 °C-pressed cathodes may permit delayed activation of electrochemically isolated sulfur domains, gradually accessed over cycling and contributing to apparent Coulombic efficiency values above 100%. Following statements have been added to the revised manuscript and Coulombic efficiency data of the rate performance test results have been added to Fig. 4 and Figs. S20 and S21.

- In Results and Discussion section of the revised Manuscript:

A gradual decline in Coulombic efficiency was observed during extended cycling (Fig. 4b), particularly in the RT-pressed and slurry-cast samples, indicating increasing interfacial resistance.

- In Results and Discussion section of the revised Manuscript:

While consistently delivering higher capacities across all C-rates in the rate performance test and at 0.1 C during cycling, the 80 °C-pressed cathodes (5 mg cm⁻²) exhibited Coulombic efficiency values exceeding 100%, likely due to delayed activation of isolated sulfur domains within the sulfur cathodes, unlike the consistently lower Coulombic efficiency observed in the RT-pressed and slurry-cast S-C cathodes. Notably, EIS analysis for the 1 and 5 mg cm⁻² electrodes revealed trends consistent with those observed at 3 mg cm⁻², reinforcing the generality of the impedance behavior across various loadings (Fig. S20, S21, Table S9).

- In the revised Manuscript:

Figure 4. Electrochemical performance of RT-pressed and 80 °C-pressed S-C composite cathodes. a) Rate capability test results and galvanostatic charge-discharge cycling results of the RT- and 80 °C-pressed, and slurry cast S-C composite cathodes b) areal mass loading of 3 mg_s cm⁻² at 0.2 C. Voltage profiles of c) RT-pressed, d) 80 °C-pressed, and e) slurry-cast S-C composite cathodes (1 C = 1675 mA g⁻¹). CV results of the f) RT-pressed, g) 80 °C-pressed h) slurry-cast S-C composite cathodes. Nyquist plot of i) RT-pressed, j) 80 °C-pressed, and k) slurry-cast S-C composite cathodes, **measured at the fully charged state** before and after 30 charge-discharge cycling and **l) equivalent circuit model**. Galvanostatic charge-discharge cycling results of the RT- and 80 °C-pressed, and slurry cast S-C composite cathodes **m)** areal mass loading of 1 mg_s cm⁻² at 1.0 C.

- In the revised Supplementary Information:

Figure S20. Electrochemical performance of the slurry cast S-C composite cathodes with areal mass loading of $1 \text{ mg}_S \text{ cm}^{-2}$. a) Rate capability test results, voltage profiles of b) RT-pressed, c) 80 °C-pressed, and d) slurry-cast S-C composite cathodes at various C-rate ($1 \text{ C} = 1675 \text{ mAh gs}^{-1}$) and Nyquist plot of e) RT-pressed, f) 80 °C-pressed, and g) slurry-cast S-C composite cathodes, **measured at the fully charged state** before and after 30 charge-discharge cycling.

Figure S21. Electrochemical performance of the slurry cast S-C composite cathodes with areal mass loading of $5 \text{ mg}_s \text{ cm}^{-2}$. a) Rate capability test results, b) cycle performance comparison, voltage profiles of c) RT-pressed, d) 80°C -pressed, and e) slurry-cast S-C composite cathodes at various C-rate ($1 \text{ C} = 1675 \text{ mAh g}_s^{-1}$) and Nyquist plot of f) RT-pressed, g) 80°C -pressed, and h) slurry-cast S-C composite cathodes, **measured at the fully charged state** before and after 50 charge-discharge cycling.

Comment 2-3: In the electrochemical analysis, the electrochemical impedance analysis of the heat-roll pressed cathode is suggested to be analyzed with an equivalent circuit model to support the corresponding discussion. The raw experimental data and the fitting data are suggested to be reported together as experimental data points and the fitting curves. [Suggestion] Please analyze the impedance data.

Response: We thank the reviewer for pointing out the lack of detailed analysis of the electrochemical impedance data. In response, we have revised the manuscript to include a comprehensive impedance fitting analysis based on an equivalent circuit model. The revised version presents both the experimental Nyquist plots and the corresponding fitted curves to support the interpretation more rigorously. In addition, the fitted values for each circuit element, along with the complete EIS fitting plots, have been provided in Table S9 in the updated Supplementary Information.

- In Results and Discussion section of the revised Manuscript:

EIS analysis performed at the charged state after 30 cycles (Figs. 4i–4k and Table S9) revealed distinct trends for each electrode type. The 80°C -pressed cathode maintained low impedance,

while both the RT-pressed and slurry-cast electrodes exhibited notable increases in ohmic resistance (R_s). The rise in R_s for the RT-pressed electrode suggests compromised structural integrity, likely due to mechanical degradation during cycling,[43–45] whereas the thermally compressed 80 °C-pressed electrode retained a stable architecture. Similarly, the increase in R_s for the slurry-cast electrode is consistent with previous reports that conventional slurry processing can lead to solvent-induced inhomogeneity and poor interfacial stability issues.[46–48]

- In Results and Discussion section of the revised Manuscript:

Notably, EIS analysis for the 1 and 5 mg cm^{-2} electrodes revealed trends consistent with those observed at 3 mg cm^{-2} , reinforcing the generality of the impedance behavior across various loadings (Figs. S20 and S21, Table S9).

- In Results and Discussion section of the revised Manuscript:

Figure 4. Electrochemical performance of RT-pressed and 80 °C-pressed S-C composite cathodes. a) Rate capability test results and galvanostatic charge-discharge cycling results of the RT- and 80 °C-pressed, and slurry cast S-C composite cathodes b) areal mass loading of 3 mg_s cm⁻² at 0.2 C. Voltage profiles of c) RT-pressed, d) 80 °C-pressed, and e) slurry-cast S-C composite cathodes (1 C = 1675 mA g⁻¹). CV results of the f) RT-pressed, g) 80 °C-pressed h) slurry-cast S-C composite cathodes. Nyquist plot of i) RT-pressed, j) 80 °C-pressed, and k) slurry-cast S-C composite cathodes before and after 30 charge-discharge cycling **and l) equivalent circuit model**. Galvanostatic charge-discharge cycling results of the RT- and 80 °C-pressed, and slurry cast S-C composite cathodes **m)** areal mass loading of 1 mg_s cm⁻² at 1.0 C.

- In References section the revised Manuscript:

45. Lateef, S. *et al.* Understanding the effects of binder dissolution dynamics on the chemistry and performance of lithium-sulfur batteries. *EES Batteries*, **1**, 947–963 (2025).

46. Horst, M. *et al.* A binder-free dry coating process for high sulfur loading cathodes of Li-S batteries: A proof-of-concept. *J. Power Sources* **587**, 233675 (2023).

47. Fiedler, M. *et al.* Mechanistic insights into the cycling behavior of sulfur dry-film cathodes. *Adv. Sustainable Syst.* **7**, 2200439 (2023).

48. Sul, H., Lee, D. & Manthiram, A. High-loading lithium-sulfur batteries with solvent-free dry-electrode processing. *Small* **20**, 2400728 (2024)

- In the revised Supplementary Information:

Figure S20. Electrochemical performance of the slurry cast S-C composite cathodes with areal mass loading of $1 \text{ mg}_s \text{ cm}^{-2}$. a) Rate capability test results, voltage profiles of b) RT-pressed, c) 80°C -pressed, and d) slurry-cast S-C composite cathodes at various C-rate ($1 \text{ C} = 1675 \text{ mAh g}^{-1}$) and Nyquist plot of e) RT-pressed, f) 80°C -pressed, and g) slurry-cast S-C composite cathodes, **measured at the fully charged state** before and after 30 charge-discharge cycling.

Figure S21. Electrochemical performance of the slurry cast S-C composite cathodes with areal mass loading of $5 \text{ mg}_s \text{ cm}^{-2}$. a) Rate capability test results, b) cycle performance comparison, voltage profiles of c) RT-pressed, d) 80°C -pressed, and e) slurry-cast S-C composite cathodes at various C-rate ($1 \text{ C} = 1675 \text{ mAh g}^{-1}$) and Nyquist plot of f) RT-pressed, g) 80°C -pressed, and h) slurry-cast S-C composite cathodes, **measured at the fully charged state** before and after 50 charge-discharge cycling.

Table S9. Fitted parameters extracted from the equivalent circuit model shown in Fig. 4I, corresponding to the Nyquist plots in Fig. 4i–k, Fig. S20e–g, and Fig. S21f–h.

Cathodes	Areal mass ($\text{mg}_s \text{ cm}^{-2}$) & Cell state	R_s (Ω)	CPE_f (μF)	R_f (Ω)	CPE_{ct} (F)	R_{ct} (Ω)	W_o (Ω $\text{s}^{-1/2}$)
RT- pressed	1.0, before	1.99	2.04	59.32	4.22×10^{-2}	3.10	70.2
	1.0, after	1.25	9.17	46.3	1.17×10^{-2}	12.1	17.5
	3.0, before	4.16	2.11	33.0	1.64×10^{-1}	12.1	35.4
	3.0, after	42.1	1.73	24.7	7.16×10^{-4}	21.5	3.61

	5.0, before	2.47	2.28	37.3	3.45×10^{-2}	3.68	11.0
	5.0, after	19.5	3.14	22.4	3.60×10^{-1}	44.1	10.9
	1.0, before	2.08	2.08	44.7	9.53×10^{-3}	0.16	35.7
	1.0, after	2.26	2.31	14.5	3.99×10^{-4}	1.97	3.70
80 °C- pressed	3.0, before	2.25	3.82	30.6	1.48×10^{-2}	1.08	21.48
	3.0, after	5.24	3.35	8.40	3.32×10^{-6}	8.47	2.23
	5.0, before	1.24	2.51	36.6	7.56×10^{-3}	7.13	21.7
	5.0, after	16.5	2.78	20.9	1.75×10^{-4}	35.26	2.53
	1.0, before	7.90	1.99	60.1	3.88×10^{-2}	3.64	8.48
	1.0, after	19.4	1.95	110.2	3.00×10^{-2}	64.2	9.31
Slurry- cast	3.0, before	5.76	3.37	34.5	1.20×10^{-4}	7.13	1.70
	3.0, after	45.41	2.37	35.1	9.20×10^{-4}	2.64	7.86
	5.0, before	8.83	2.92	28.6	3.66×10^{-2}	8.98	10.1
	5.0, after	49.6	2.81	48.5	6.14×10^{-2}	36.41	3.11

Comment 2-4: The heat-roll pressed cathode is reported with the sulfur loadings of 1 to 5 mg cm⁻². The cell-fabrication parameters of 5 mg cm⁻² cathode cannot be found in the manuscript. The corresponding sulfur loading and content and the related electrolyte-to-sulfur ratio are suggested to be reported together for a reference and for a comparison. The amount of lithium is suggested to be reported. [Suggestion] Please report on the necessary cell-fabrication parameters.

Response: Thank you for pointing this out. We would like to clarify that the use of lithium foil with a thickness of 120 μm, as stated in the Experimental Section of the original manuscript, applies to all cells regardless of sulfur mass loading. In the revised manuscript, we have additionally updated the following sentence (highlighted in yellow) to include the E/S ratio for the 5 mg S cm⁻² electrode, which was previously omitted.

- In Method section of the revised manuscript:

The electrolyte-to-sulfur ratio (E/S) was set to 10 for 1 mgS cm⁻² electrodes, 7 for 3 mgS cm⁻² and 5 for 5 mgS cm⁻² electrodes.

Comment 2-5: After reviewing the introduction and references, it is suggested to provide a more comprehensive literature survey. Several recent studies have reported cathode fabrication methods involving heat and pressing treatments, many of which are closely related to the present work. Incorporating a discussion of these studies could help position this research within the current landscape and highlight its novelty. [Suggestion] Please consider expanding the introduction and

discussion sections to include recent advances in cathode design using thermal and pressure-assisted treatments.

Response: We appreciate the reviewer's insightful suggestion to expand the introduction and discussion with additional literature references on dry processing techniques and recent advances in Li/S batteries. In response, we have revised the manuscript to include several recent review articles that provide a comprehensive overview of current progress and remaining challenges in these areas. Furthermore, we have added citations to recent original research studies on dry-coated sulfur cathodes to contextualize our findings within the broader field. These additions are reflected in the updated manuscript.

- In the Reference section of the revised manuscript:

21. Kim, N.-Y. *et al.* Material challenges facing scalable dry-processable battery electrodes. *ACS Energy Lett.* **9**, 5688–5703 (2024).

22. Jin, W. *et al.* Advancements in dry electrode technologies: Towards sustainable and efficient battery manufacturing. *ChemElectroChem* **11**, e202400288 (2024).

23. Hong, T. H., Kim, D. J., Ko, S. M. & Lee, J. T. Solvent-free dry-process for developing high-performance lithium-sulfur batteries. *Korean J. Chem. Eng.* **42**, 1475–1490 (2025).

24. Park, J. *et al.* Sustainable and cost-effective electrode manufacturing for advanced lithium batteries: the roll-to-roll dry coating process. *Chem. Sci.* **16**, 6598–6619 (2025).

- In the revised Supplementary Information

[S14] Horst, M. *et al.* A binder-free dry coating process for high sulfur loading cathodes of Li-S batteries: A proof-of-concept. *J. Power Sources* **587**, 233675 (2023)

[S15] Hu, J.-K. *et al.* Dry electrode technology for scalable and flexible high-energy sulfur cathodes in all-solid-state lithium-sulfur batteries. *J. Energy Chem.* **71**, 612–618 (2022).

[S16] Fiedler, M. *et al.* Mechanistic insights into the cycling behavior of sulfur dry-film cathodes. *Adv. Sustainable Syst.* **7**, 2200439 (2023).

[S17] Sul, H., Lee, D. & Manthiram, A. High-loading lithium-sulfur batteries with solvent-free dry-electrode processing. *Small* **20**, 2400728 (2024).

Thank you very much for the valuable comments. Your comments significantly improved the quality of the manuscript.

Reviewer #3

Overall comment: This manuscript presents a solvent- and binder-free approach for fabricating sulfur-carbon composite cathodes via thermally assisted dry-pressing, leveraging sulfur's softening behavior to serve as an intrinsic binder. The experimental work is well executed and supported by structural characterizations, in-situ observations, and mechanical simulations. The study addresses a practical and timely issue in green and scalable manufacturing of Li–S batteries,

achieving long cycle life and competitive capacity metrics. However, the conceptual novelty of the work is limited. The central idea is largely based on the well-known thermal softening behavior of sulfur and does not introduce fundamentally new mechanisms in materials design or electrochemistry. Moreover, the role of sulfur as a persistent structural binder under cycling conditions is not convincingly demonstrated. The manuscript also lacks a clear differentiation from existing binder-free cathode strategies. In its current form, the work does not meet the originality and mechanistic depth expected for publication in Nature Communications.

Response: We sincerely thank the reviewer for the thoughtful and constructive feedback. We have carefully addressed all comments and revised the manuscript accordingly. We hope that the revisions meet the reviewer's expectations. Point-by-point responses to each comment are provided below.

Comment 3-1: The manuscript lacks experimental or theoretical validation that sulfur retains binding capability after being converted into soluble polysulfides.

Response: We thank the reviewer for this important comment. To address it, we performed ex-situ SEM imaging on electrodes at 40% depth of discharge (DOD), as shown in the revised Supplementary Fig. S23. These images demonstrate that the 80 °C-pressed electrodes retain a mechanically cohesive framework during the early stages of electrochemical cycling, while the RT-pressed counterparts already exhibit signs of surface deformation and localized cracking. We suggest that not all sulfur in the electrode undergoes electrochemical conversion simultaneously. As discussed in the literature, including [Wujcik et al. J. Electrochem. Soc., 164, A18 (2017)], electrochemical and chemical reactions may proceed heterogeneously across the electrode, with some regions forming solid Li_2S_2 or Li_2S while other areas still contain unreacted sulfur or soluble polysulfides. This spatial nonuniformity helps explain the coexistence of partially converted sulfur species and maintained structural integrity in localized domains, as captured in our SEM observations. We have revised the manuscript to include this new data and discussion. The added content has been highlighted in yellow in the revised version.

- In Results and Discussion section of the revised Manuscript:

Additional ex-situ SEM images collected at ~40% depth of discharge at the initial discharge process reveal that the 80 °C-pressed cathodes retain structural cohesion during early electrochemical cycling (Fig. S23). This observation, together with prior findings that sulfur conversion does not occur uniformly across the electrode,[52] suggests that partially unreacted sulfur and early-stage Li_2S_x species can coexist, allowing localized structural retention during the formation of discharge products. Importantly, the observations suggest that the initial mechanical integrity provided by heat-pressing plays a critical role in mitigating the potential structural weakening associated with the solid-liquid-solid transition of sulfur during cycling. Although polysulfide formation and subsequent sulfur precipitation inherently involve relocation of active material, the 80 °C-pressed cathode

preserves a cohesive framework that accommodates these transformations without particle detachment or severe cracking, as confirmed by ex-situ SEM.

- In the revised Supplementary Information:

Figure S23. Ex-situ SEM images of S-C cathodes collected at ~40% depth of discharge at the initial discharge process. (a) RT-pressed cathode, (b) 80 °C-pressed cathode.

- In Reference section of the revised Manuscript:

52. Wujcik, K. H. *et al.* In situ X-ray absorption spectroscopy studies of discharge reactions in a thick cathode of a lithium sulfur battery. *J. Electrochem. Soc.* **164**, A18–A27 (2017).

- In Methods section of the revised Manuscript:

Ex-situ SEM was performed on electrodes obtained at approximately 40% DOD during the initial discharge cycle or after five charge-discharge cycles. For 40 % DOD cathode, coin-type half-cells were disassembled inside an Ar-filled glove box after reaching the targeted discharge capacity. The cathodes were gently rinsed with anhydrous DOL/DME mixture solution to remove residual electrolyte and lithium salts, followed by vacuum drying at RT for 2 hours. Surface morphology was examined using a field-emission scanning electron microscope (Auriga Zeiss) operated at 5 kV accelerating voltage.

Comment 3-2: While ex-situ images demonstrate morphological integrity, they do not adequately explain how the electrode resists internal fracture or delamination under repeated volume changes.

Response: We thank the reviewer for this insightful comment. In response, we have revised the manuscript to further elaborate on the mechanisms by which the 80 °C-pressed cathodes resist internal fracture and delamination during repeated volume changes. A discussion addressing these points has been added to the revised manuscript.

- In Results and Discussion section of the revised Manuscript:

These collective findings of ex-situ SEM and S/Li₂S symmetric cell test highlight that the improved morphological integrity of the 80 °C-pressed cathodes not only supports structural robustness but also accommodates the chemo-mechanical stresses induced by repeated sulfur redox cycling. Given that sulfur serves both as the active material and a structural binder in this solvent- and binder-free system, its homogeneous distribution

within the carbon matrix is particularly critical. The resulting uniform composite architecture enhances interparticle cohesion and mitigates fracture propagation during solid–liquid–solid transitions. This structural coherence is essential to maintaining continuous electron and ion transport pathways, preventing the detachment of active material, and ultimately enabling prolonged cycling stability.

Comment 3-3: Comparison with prior binder-free strategies is insufficient. Previous studies using melt-infiltration, porous carbons, or 3D scaffolds have pursued similar goals.

Response: We thank the reviewer for the valuable suggestion. To better contextualize our approach, we have revised the manuscript to emphasize the differences between our method and previous binder-free strategies that utilize melt-infiltration, porous carbon frameworks, or 3D scaffolds. Specifically, we highlight that many prior approaches rely on nonstandard current collectors and processing methods that are not compatible with scalable manufacturing, such as roll-to-roll coating, and often involve high-temperature or complex fabrication steps that increase production costs. In contrast, our method employs a conventional aluminum foil current collector and requires no solvents or binders, offering a more practical and cost-effective future direction toward scalable Li–S battery manufacturing. The following statement has been added to the revised manuscript to reflect this comparison.

- In Introduction section of the revised Manuscript:

However, prior binder-free strategies often rely on specialized or nonstandard current collectors that are incompatible with scalable manufacturing methods such as roll-to-roll coating, ultimately increasing production costs. In some cases, high-temperature melt-infiltration of sulfur into carbon substrates is also required,[29] adding to process complexity and material expenses. Moreover, these approaches typically involve rigid or pre-structured porous carbon architectures where precise control over bulk porosity is difficult to achieve. These limitations make it challenging to implement such methods in high-throughput, cost-sensitive manufacturing environments, thereby restricting their scalability.

- In Reference section of the revised Manuscript:

29. Um, K. *et al.* Janus architecture host electrode for mitigating lithium-ion polarization in high-energy-density Li-S full cells. *Energy Environ. Sci.* **17**, 9112–9121 (2024).

Comment 3-4: There is no quantitative modeling or correlation to electrochemical performance metrics such as impedance, diffusion kinetics, or polarization behavior. This weakens the causal link between electrode structure and performance.

Response: We appreciate the comment pointing out the need for deeper interpretation of the impedance data. We have revised the manuscript to include a comprehensive impedance fitting

analysis based on an equivalent circuit model. The revised version presents both the experimental Nyquist plots and the corresponding fitted curves to support the interpretation more rigorously. In addition, the fitted values for each circuit element, along with the complete EIS fitting plots, have been provided in Table S9 in the updated Supplementary Information.

- In Results and Discussion section of the revised Manuscript:

EIS analysis performed at the charged state after 30 cycles (Fig. 4i–4k and Table S9) revealed distinct trends for each electrode type. The 80 °C-pressed cathode maintained low impedance, while both the RT-pressed and slurry-cast electrodes exhibited notable increases in ohmic resistance (R_s). The rise in R_s for the RT-pressed electrode suggests compromised structural integrity, likely due to mechanical degradation during cycling,[43–45] whereas the thermally compressed 80 °C-pressed electrode retained a stable architecture. Similarly, the increase in R_s for the slurry-cast electrode is consistent with previous reports that conventional slurry processing can lead to solvent-induced inhomogeneity and poor interfacial stability issues.[46–48]

- In Results and Discussion section of the revised Manuscript:

Notably, EIS analysis for the 1 and 5 mg cm⁻² electrodes revealed trends consistent with those observed at 3 mg cm⁻², reinforcing the generality of the impedance behavior across various loadings (Fig. S20, S21, Table S9).

- In the revised Supplementary Information:

Table S9. Fitted parameters extracted from the equivalent circuit model shown in Fig. 4l, corresponding to the Nyquist plots in Fig. 4i–k, Fig. S20e–g, and Fig. S21f–h.

Cathodes	Areal mass (mgs cm ⁻²) & Cell state	R_s (Ω)	CPE_f (μ F)	R_f (Ω)	CPE_{ct} (F)	R_{ct} (Ω)	W_o (Ω s ^{-1/2})
RT- pressed	1.0, before	1.99	2.04	59.32	4.22×10^{-2}	3.10	70.2
	1.0, after	1.25	9.17	46.3	1.17×10^{-2}	12.1	17.5
	3.0, before	4.16	2.11	33.0	1.64×10^{-1}	12.1	35.4
	3.0, after	42.1	1.73	24.7	7.16×10^{-4}	21.5	3.61
	5.0, before	2.47	2.28	37.3	3.45×10^{-2}	3.68	11.0
	5.0, after	19.5	3.14	22.4	3.60×10^{-1}	44.1	10.9
80 °C- pressed	1.0, before	2.08	2.08	44.7	9.53×10^{-3}	0.16	35.7
	1.0, after	2.26	2.31	14.5	3.99×10^{-4}	1.97	3.70
	3.0, before	2.25	3.82	30.6	1.48×10^{-2}	1.08	21.48
	3.0, after	5.24	3.35	8.40	3.32×10^{-6}	8.47	2.23

	5.0, before	1.24	2.51	36.6	7.56×10^{-3}	7.13	21.7
	5.0, after	16.5	2.78	20.9	1.75×10^{-4}	35.26	2.53
	1.0, before	7.90	1.99	60.1	3.88×10^{-2}	3.64	8.48
	1.0, after	19.4	1.95	110.2	3.00×10^{-2}	64.2	9.31
Slurry-cast	3.0, before	5.76	3.37	34.5	1.20×10^{-4}	7.13	1.70
	3.0, after	45.41	2.37	35.1	9.20×10^{-4}	2.64	7.86
	5.0, before	8.83	2.92	28.6	3.66×10^{-2}	8.98	10.1
	5.0, after	49.6	2.81	48.5	6.14×10^{-2}	36.41	3.11

Comment 3-5: The manuscript emphasizes the industrial potential of the method but omits key details such as roll-to-roll process compatibility, throughput, or large-area uniformity. More discussion or preliminary validation would be necessary.

Response: We thank the reviewer for emphasizing the importance of validating the industrial relevance of the proposed method. As noted in our response to Comment 3-3, we have clarified in the revised manuscript that our binder- and solvent-free process is fully compatible with conventional aluminum foil current collectors and roll press-based fabrication, making it inherently transferable to industrial roll-to-roll platforms and well-suited for scalable production in the future. Furthermore, the manuscript demonstrates that the process allows control over bulk porosity by adjusting the roll-gap for different pressing temperature, which serves as a key processing variable. Such tunable control over electrode porosity is not readily achievable in the previously mentioned approaches, particularly those based on three-dimensional porous carbon substrates, where the inherent structural rigidity and irregular morphology limit precise control over electrode densification and pore structure. This capability enables meaningful comparison of electrodes fabricated under different thermal conditions and underscores the tunability and scalability of the method. The following statement has been added to the revised manuscript to reflect this comparison.

- In Introduction section of the revised Manuscript:

The method is based on dry powder formation on Al foil followed by roll-pressing, both of which are compatible with continuous processing and allow control over key processing parameters such as areal loading and electrode porosity, highlighting its potential for scalable and tunable manufacturing.

Comment 3-6: The SEM image in Figure 2 is difficult to interpret. It is recommended to include both a magnified view and a lower-magnification overview to aid in visual comprehension of the particle morphology and packing behavior.

Response: We appreciate the reviewer's helpful suggestion. In response, higher-magnification SEM images have been added to the Supplementary Information (Fig. S10) to better illustrate

particle morphology and packing characteristics. These images provide clearer contrast between the surface structure of the RT- and 80 °C-pressed composites and support the structural discussion in the main text. The following statement has been added to the revised manuscript and the higher magnification images have been added to the supplementary information.

- In Results and Discussion section of the revised Manuscript:

Higher-magnification SEM images (Fig. S10) show that RT-pressed samples exhibit uneven sulfur distribution with particles remaining outside aggregates, while 80 °C-pressed samples exhibit more uniform morphology and sulfur distribution.

- In the revised Supplementary Information:

Figure S10. High magnification SEM images of S-C composite after the compression at a) RT and b) 80 °C.

Thank you very much for the valuable comments. Your comments significantly improved the quality of the manuscript.

Reviewer #4

Overall comment: This work presents a new way to fabricate solvent-free Li/S batteries with reduced fabrication costs and improved performance. The work is interesting and well written. However, to be accepted by Nature Communications, the following issues should be considered.

Response: We sincerely thank the reviewer for the positive and encouraging feedback on our manuscript. In response to the concerns raised, we have carefully revised the manuscript and addressed each point in detail. We hope that these revisions sufficiently clarify the issues and strengthen the manuscript. Point-by-point responses are provided below.

Comment 4-1: Figure 1 b, clarify the procedure you used to characterize the powder rheology – usually this is performed by shear cell, or FT4.

Response: We thank the reviewer for the comment. A clarifying statement describing the shear cell-based method used for the elemental sulfur and S–C composite powder rheology test has been added to the revised manuscript.

- In Method section of the revised Manuscript:

Powder rheological properties were evaluated using a powder shear cell tester (TA Instruments, Discovery HR-20). Over 20 mL of powder was loaded into a stainless steel cylindrical cell (28 mm diameter). The assembly was then heated to the target temperature—either RT or 80 °C—at a ramp rate of 3 °C min⁻¹. Upon reaching the set temperature, the powder was equilibrated for 5 minutes to ensure thermal uniformity throughout the sample. A pre-consolidation pressure of 9 kPa was applied to condition the powder bed and achieve a uniform packing state. Excess powder was removed after consolidation, resulting in an exact sample volume of 18 mL for shear testing. The sample was then subjected to a constant shear rate (0.001 rad s⁻¹) under fixed normal stress until shear failure occurred.[53] The normal stress was systematically decreased to 7, 6, 5, 4, and 3 kPa in successive cycles, following instrument protocols. A yield locus was constructed by plotting the steady-state shear stress versus normal stress, and a linear fit was applied. Using Mohr's circle analysis, the unconfined yield strength (UYS) and the major principal stress (MPS) were extracted: (1) The first Mohr's circle was constructed with one point at the origin (0,0) and a second point tangent to the yield locus; its x-axis intersection yields the UYS. (2) The second Mohr's circle used the pre-shear point and a tangent to the yield locus; its x-axis intersection defines the MPS[54–56] The flow function (FF) of the powder was calculated using the UYS and MPS values, as expressed in Equation shown below.

$$FF = \frac{MPS}{UYS}$$

- In Methods section of the revised Manuscript:

The same shear cell-based method described above was employed to characterize the rheological behavior of the S–C composite powder under identical testing conditions.

- In References section of the revised Manuscript:

53. Krantz, M., Zhang, H. & Zhu, J. Characterization of powder flow: Static and dynamic testing. *Powder Technol.* **194**, 239–245 (2009).

54. Rhodes, M. *Storage and flow of powders-hopper design*. In *introduction to particle technology 2nd edn* 265–292 (John Wiley & Sons, Chichester, 2008) <https://doi.org/10.1002/9780470727102.ch10>.

55. Guerin, E. *et al.* Rheological characterization of pharmaceutical powders using tap testing, shear cell and mercury porosimeter. *Int. J. Pharm.* **189**, 91–103 (1999).

56. Macri, D. *et al.* Characterization of the bulk flow properties of industrial powders from shear tests. *Processes* **8**, 540 (2020).

Comment 4-2: Line 125, clarify if you are using 50 mm powder bed - This seems strange to describe the thin electrode behaviour.

Response: We thank the reviewer for the comment. The 50 mm powder bed mentioned in the manuscript refers to a parameter used in the sulfur powder simulation and is not related to the physical thickness of any fabricated electrode. This simulation domain size was selected to ensure that at least ten sulfur particles with a representative diameter of ~5 μm could be included in each spatial direction, thereby allowing for statistically meaningful assessment of

bulk powder behavior. The purpose of this simulation was to examine the intrinsic morphological response and packing characteristics of elemental sulfur under idealized conditions, rather than to replicate the geometry of thin-film electrodes. The following statement has been to the revised manuscript.

- In Methods section of the revised Manuscript:

The simulation domain was sized to include at least ten sulfur particles with a representative diameter of $\sim 5 \mu\text{m}$ in each spatial direction, ensuring statistically meaningful assessment of bulk powder behavior.

Comment 4-3: 3. Figure 1 g-j, (1) Are you using ROCKY to quantify the particle deformation? please clarify if FEM approach is used in ROCKY module, if yes, can you extract the pressure acting on the punch? to calibrate your simulation results. (2) need to specify the simulation parameters and calibration procedure of bond model used for the simulation. (3) the spherical particle approximation may not be suitable to describe sulphur particles. (4) Line 407-408, where did you find the general relationships of particle property and temperature. In general, this part needs more experimental/theoretical support to make sure your finding sound enough!

Response: Thank you for your valuable feedback on our manuscript.

(1) We clarify that ROCKY was not employed for this purpose. The primary objective of our simulation was to qualitatively evaluate the influence of material properties at room temperature (RT) and 80°C on the compactness of the powder bed, as observed in our experimental results (see Fig. 1c and Fig. 1f). Given the unavailability of material properties required for compaction simulations at various temperatures, this approach was the most practical and effective choice for our study. Regarding the simulation methodology, we utilized ANSYS Workbench, which integrates both ROCKY and Mechanical modules. The finite element method (FEM) was applied exclusively within the Mechanical module, not within ROCKY. As the focus of our study was qualitative analysis rather than quantitative comparison, we did not pursue calibration of the simulation results or extraction of specific parameters such as the pressure acting on the punch. To clarify this, we revised the statement in the manuscript.

- In Results and Discussion section of the revised Manuscript:

The simulation provides insights into particle-level deformation and densification under thermal and mechanical stresses, illustrating how elevated temperatures influence sulfur particle rearrangement, deformation, and packing efficiency. Noted that the simulation is designed for qualitative insights rather than quantitative comparison with experimental data.

- In Methods section of the revised Manuscript: The roll-pressing of sulfur powder bed is modeled using ANSYS Workbench, a commercial software package that integrates ROCKY and Mechanical modules,

(2) In this study, we employed the cohesive bond model available in ANSYS Rocky to simulate interparticle forces during the compaction of sulfur powder. The model parameters were selected to represent the weakly cohesive and brittle nature of sulfur particles. Considering the sulfur particle diameter in the model ($d_i = 5\mu\text{m}$), a maximum elongation of 1 μm was specified, allowing bond breakage when significant separation occurs between particles. The damping ratio was set to 0.25 to account for energy dissipation and enhance numerical stability. A distance factor of 0.1 was used to ensure that bonds form only between closely packed particles, mimicking realistic particle contact conditions. The normal and tangential stiffness per unit area were both set to 100 kN/m^2 to provide moderate resistance to deformation while maintaining computational efficiency. The tensile and shear stress limits were each set to 10 kPa , reflecting the low bonding strength typical of sulfur powder. Additionally, a bond activation time of 0.1 seconds was specified to allow particle rearrangement before bond formation, representing delayed cohesion during compaction. These values were chosen to approximate the mechanical behavior of loosely agglomerated particles and allow for realistic fracture and rearrangement during compaction. While no universally accepted bond model parameters exist for sulfur powder, the selected values fall within commonly reported ranges for weakly cohesive granular materials and provide a practical basis for capturing the essential mechanics of sulfur particle compaction. Further refinement and validation can be performed by calibrating against experimental data such as stress-strain response or final compact density.

We did not perform a calibration of the bond model, as it was not essential for the primary objective of this simulation. The goal was to investigate how different material properties at RT and 80 °C influence the compactness of the powder bed, as qualitatively observed in our compaction experiments (see Fig. 1c and Fig. 1f). While we acknowledge that a quantitative analysis and a detailed comparison between simulation and experimental results would require calibration of the bond model, such an effort was beyond the scope of the present study. To clarify this, we revised statement relevant to this in the revised manuscript.

- In Results and Discussion section of the revised Manuscript: **To understand the pronounced morphological transformation observed at 80 °C**, the thermal compression behavior of elemental sulfur particles under conditions mimicking the roll-pressing process was simulated.
- In Methods section of the revised Manuscript: The objective of the model is to **qualitatively** understand how temperature changes in the roll-pressing process affect the sizes and shapes of particles in the resulting hot-rolled sintered film.
- In Methods section of the revised Manuscript: **The bond model parameters used in the simulation include a maximum elongation of 1 μm , a damping ratio of 0.25, a distance factor of 0.1, normal and tangential stiffness per unit area of 100 kN/m^2 , tensile and shear stress limits of 10 kPa , and a bond activation time of 0.1 seconds. These values were selected to approximate the mechanical behavior of loosely agglomerated particles and to enable realistic fracture and rearrangement during compaction.**

(3) To represent sulfur particle's geometry, this study modeled the particles as spheropolygons with 12 vertices to capture irregular, faceted particle geometries and to allow the computation of inter-particle contact mechanics and packing behavior. To clarify this, the relevant statement has been revised in the revised manuscript.

- In Method section of the revised Manuscript:

The computational domain, which measures $50 \times 50 \times 50 \text{ mm}^3$, contains polygonal sulfur powder particles modeled as spheropolygons with 12 vertices, an initial equivalent spherical diameter $d_i = 5 \text{ mm}$, and a vertical-to-horizontal aspect ratio of 1. This spheropolygon representation captures the irregular, faceted geometry of actual particles while enabling computation of inter-particle contact mechanics and packing behavior.

(4) Due to the lack of experimentally measured temperature-dependent mechanical properties for sulfur particles, we adopted a simplified linear model for the temperature dependence of Young's modulus, which is commonly applicable to crystalline solids below their phase transition temperatures [Courtney, T. H. "Mechanical Behavior of Materials." 2nd ed., McGraw-Hill, 2000. ISBN: 978-0070132658]. The relationship is given by $E/E_0 = [1 - a(T/T_m)]$, where E is the modulus at a temperature T , E_0 is the modulus at 0 K, T_m is the melting point, and a is a material-specific temperature coefficient. For many crystalline solids, a is on the order of 0.5 [Courtney, T. H. "Mechanical Behavior of Materials." 2nd ed., McGraw-Hill, 2000. ISBN: 978-0070132658]. In this study, we assumed a higher value of $a = 1$ considering the particle-form of materials and to ensure a significantly reduced modulus at 80°C . With these assumptions, the linear model yielded estimated modulus of approximately 5 GPa. The slightly overestimated temperature coefficient led to a notable difference in particle compaction behavior compared to simulations at 25°C , aligning with our experimental observations. The statement relevant to this discussion has been revised and the literature has been cited in the revised manuscript.

- In Methods section of the revised Manuscript:

Due to the limited data for sulfur particles at various temperatures, we assume $E = 10 \text{ GPa}$ [59] and $\nu = 0.2$ at 25°C . Most crystalline materials possess ν in the range of 0.2 to 0.3 at 25°C . [60] At 80°C , we assume $E = 5 \text{ GPa}$ and $\nu = 0.3$, based on the linear model for the temperature dependence of Young's modulus. [61]

- In References section of the revised Manuscript:

61. Courtney, T. H. Mechanical behavior of materials (Waveland Press, 2000)

Comment 4-4: Figure 4, will the compressing level significantly affect the battery performance?

Response: We thank the reviewer for the comment. In this work, the bulk porosity was deliberately set to approximately 55%, a level previously shown to provide an optimal balance

between electrolyte accessibility and ionic/electronic transport in sulfur cathodes. This choice aligns with earlier findings, including previous study published in Nature Communications [Kang et al., Nat. Commun. 10, 3366 (2019)], which demonstrated that cathode porosity critically influences sulfur utilization and long-term cycling stability. Beyond porosity, the tortuosity of the electrode structure plays a key role in governing ion transport and reaction homogeneity. Lower tortuosity has been shown to enhance Li⁺ diffusion and promote more uniform electrochemical reactions, ultimately improving sulfur utilization and rate performance [Fu et al., Adv. Energy Mater. 13, 2300602 (2023)]. The thermomechanical pressing strategy employed here is expected to reduce tortuosity through uniform densification and improved structural alignment, potentially contributing to the enhanced cycling performance observed for the 80 °C-pressed cathodes. The following articles have been newly cited in the revised manuscript.

- In References section of the revised Manuscript:

49. Fu, Y. *et al.* Understanding of low-porosity sulfur electrode for high-energy lithium-sulfur batteries. *Adv. Energy Mater.* **13**, 2203386 (2023).
50. Kang, N. *et al.* Cathode porosity is a missing key parameter to optimize lithium-sulfur battery energy density. *Nat. Commun.* **10**, 4597 (2019).

Comment 4-5: By using the CT-scan microstructures, do the authors need to add some transport/electrochemical simulation results to indicate the mesoscale effect on the electrochemical property?

Response: We appreciate the reviewer’s insightful suggestion. We fully agree that mesoscale transport or electrochemical modeling could offer valuable insights into structure–property relationships. However, the present study is deliberately focused on experimentally validating a scalable, binder-free fabrication strategy, complemented by comprehensive structural and electrochemical characterization across multiple length scales. While transport simulations were not included in this work, we believe that the integration of such simulations with structural and electrochemical analyses will further strengthen the understanding of how electrode architecture governs performance. We anticipate that extending this work to include multiscale modeling using experimentally derived microstructures will be a promising direction for future studies to elucidate transport limitations and guide rational electrode design.

Comment 4-6: Table S1, ‘kg’ should be deleted for the amount of elemental sulfur.

Response: We thank the reviewer for pointing out the typo. The unit “kg” has been removed from Table S1 in the revised manuscript.

Thank you very much for the valuable comments. Your comments significantly improved the quality of the manuscript.

Responses to Reviewers' Comments

Responses to the reviewer's comments:

In the following pages, we have provided detailed and complete responses to the reviewer's comments and observations made to the manuscript (Research Article, No. NCOMMS-25-41503A) entitled, "Unveiling Sulfur as a Binder for High-Performance Polymer-Free Sulfur-Carbon Cathodes via Solvent-Free Fabrication". The revised or newly added contents have been highlighted in yellow in the revised Manuscript. These changes are also marked in yellow in this response to the reviewers' comments.

Reviewer #1

Overall comment: The authors have added new data. However in some cases, only literature was added. Symmetrical cathode tests are added which is valuable.

Response: We greatly appreciate the reviewer for positive comments. We hope that our revisions have addressed the reviewer's concerns. Point-to-point replies to the reviewer's comments are listed below.

Comment 1-1: Amorphous content: PDF only gives limited insight. Normally, amorphous content is determined with internal intensity standard.

Response: We appreciate the reviewer's feedback regarding the crystallographic analysis of sulfur. We performed an internal intensity standard approach by preparing a series of crystalline sulfur-Ketjen Black (S-C) reference mixtures with known sulfur ratios to calibrate the diffraction intensity. The integrated intensities of the sulfur (222), (026), and (040) reflections exhibited strong linearity ($R^2 > 0.97$) with crystalline sulfur content. Using this calibration, the S-C composite was found to contain ~44 wt.% crystalline sulfur in the total composite. Given that the composite contains ~70 wt.% total sulfur (based on TGA results), this corresponds to approximately 63 % of the sulfur existing in a crystalline form, while the remaining ~37 % is amorphous or poorly crystalline. The following statement, figures and table have been added to the revised Manuscript and Supplementary Information.

- In Results and Discussion section of the revised Manuscript:

From the integrated peak areas of XRD patterns shown in Fig. S9, the crystalline sulfur fraction in the S-C composite was estimated to be approximately 44 wt.% of the total composite, corresponding to about 63 % of the total sulfur content. This partial

amorphization suggests effective sulfur dispersion and confinement within the carbon matrix.

- In Methods section of the revised Manuscript:

To evaluate the amorphous sulfur content in the sulfur–carbon (S–C) composites, a calibration series of crystalline S–C ($S_{\text{cryst-C}}$) samples was prepared by mechanically mixing sulfur and Ketjen black at weight ratios of 90:10 (90 S_{cryst}), 70:30 (70 S_{cryst}), 50:50 (50 S_{cryst}) and 40: 60 (40 S_{cryst}) without any heat treatment. All $S_{\text{cryst-C}}$ mixtures were pre-weighed to maintain a constant carbon mass on the sample holder. For reference, pure crystalline sulfur (100 S_{cryst}), Ketjen black, and the synthesized S–C composites were also analyzed. XRD was performed using a PANalytical Aeris diffractometer with a Cu-K α source ($\lambda = 1.5406 \text{ \AA}$), scanning from 20° to 33° 2θ with a step size of 0.01° and a dwell time of 200 s per step. The integrated intensities of the sulfur (222), (026), and (040) reflections were obtained and used to construct calibration curves of integrated peak area versus crystalline sulfur content. Linear regression yielded R^2 values exceeding 0.95 for all peaks, indicating strong linearity. The relative crystalline sulfur content of the S–C composite was then determined by extrapolation from the regression model, and the amorphous sulfur fraction was estimated by subtracting the crystalline contribution from the total sulfur content. This analytical approach follows methodologies established in prior XRD-based phase quantification studies [63,64].

- In the revised Supplementary Information:

Figure S9. (a) XRD patterns of reference crystalline sulfur, crystalline sulfur–carbon mixtures ($S_{\text{cryst-C}}$) with varying sulfur ratios and the synthesized S–C composite. Major sulfur reflections are labeled. (b) Linear regression of integrated peak areas for the (222), (026), and (040) reflections versus the crystalline sulfur content, yielding R^2 values of 0.97–0.99. The estimated crystalline sulfur fraction in the S–C composite (highlighted region) is approximately 44 wt.%. XRD Peak integration results are summarized in Table S5.

- In the revised Supplementary Information:

Table S5. XRD Peak integration results of S_{cryst}-C and the S-C composites

Sample	(222)		(026)		(040)	
	FWHM (°)	Integrated area (a.u.)	FWHM (°)	Integrated area (a.u.)	FWHM (°)	Integrated area (a.u.)
100S _{cryst}	0.16	16148	0.16	8367	0.16	6662
90S _{cryst} -C	0.16	12835	0.16	6178	0.16	5174
70S _{cryst} -C	0.16	8312	0.16	4554	0.16	3402
50S _{cryst} -C	0.16	4203	0.16	2210	0.16	1935
40S _{cryst} -C	0.16	2882	0.16	1193	0.16	1590
S-C	0.17	3210	0.17	1433	0.17	1354

Comment 1-2: Benchmark: The authors should measure a benchmark electrode with liquid electrolyte for comparison themselves. A solid state battery is not a proper benchmark.

Response: Thank you for the valuable comments. We agree that a solid-state battery study is not an appropriate benchmark for our work. Accordingly, we have removed the all-solid-state reference and added two literature benchmarks that are directly comparable to our system, reporting 3D-printed or dry-cast electrodes tested with liquid electrolytes. Despite the higher active material content and the absence of binder in our electrodes, our system demonstrates superior electrochemical performance. The updated benchmarking details are provided in the revised Supplementary Information with slight reorder of the reference list.

- In the revised Supplementary Information:

[S13] Gao, X. *et al.* Toward a remarkable Li-S battery via 3D printing. *Nano Energy* **56**, 595–603 (2019).

[S14] Um, K. *et al.* Janus architecture host electrode for mitigating lithium-ion polarization in high-energy-density Li-S full cells. *Energy Environ. Sci.* **17**, 9112–9121 (2024).

[S15] Horst, M. *et al.* A binder-free dry coating process for high sulfur loading cathodes of Li-S batteries: A proof-of-concept. *J. Power Sources* **587**, 233675 (2023).

[S16] Sul, H., Lee, D. & Manthiram, A. High-loading lithium-sulfur batteries with solvent-free dry-electrode processing. *Small* **20**, 2400728 (2024).

[S17] Kim, D.J. *et al.* Solvent-free dry-process enabling high-areal loading selenium-doped SPAN cathodes toward practical lithium-sulfur batteries. *Small* 21, 2503037 (2025).

Comment 1-3: The authors should provide pouch cell data, not only refer to literature data.

Response: Thank you for the valuable suggestion. Following your advice, we fabricated pouch cells using our binder-free, dry-cast S-C cathode and evaluated the performance. The pouch cell delivered good cycling performance, demonstrating that our electrode has promising potential in a more practical format beyond coin cells. We have included the full pouch-cell procedures and data in the Supplementary information (Fig. S21) and noted these results in the revised manuscript. We appreciate the reviewer's guidance, which strengthened the manuscript.

- In Results and Discussion section of the revised Manuscript:

The 80 °C-pressed cathode was further evaluated in a pouch cell ($1.5 \times 1.5 \text{ cm}^2$, 3 mgs cm^{-2} , Fig. S21), which delivered an initial capacity of 1000 mAh g^{-1} and maintained 897 mAh g^{-1} after 30 cycles, demonstrating the applicability of the binder-free S-C cathodes to a more practical cell form factor and consistent electrochemical behavior with the corresponding coin-cell results.

- In Experimental section of the revised Manuscript:

For pouch cells, the 80 °C-pressed cathode ($1.5 \times 1.5 \text{ cm}^2$, 3 mgs cm^{-2}) was assembled together with the carbon-coated Celgard 2400, Li metal anode, and the same electrolyte as in the coin cells.

- In the revised Supplementary Information:

Figure S21. Charge discharge cycling performance of the pouch cell using 80 °C-pressed S-C composite cathode ($1.5 \times 1.5 \text{ cm}^2$, 3 mgs cm^{-2}) at 0.3 C.

Thank you very much for the valuable comments. Your comments significantly improved the quality of the manuscript.

Reviewer #2

Overall comment: The paper's quality has improved significantly since its revision. By incorporating all the suggestions, the authors produced a revised manuscript that is now recommended for publication.

Response: We again sincerely thank the reviewer for the constructive and encouraging feedback. We believe the manuscript has been improved substantially after the revision.

Thank you very much for the valuable comments. Your comments significantly improved the quality of the manuscript.

Reviewer #3

Overall comment: The authors have carefully addressed the earlier concerns by adding new experimental data (ex-situ SEM, EIS fitting, high-magnification SEM) and expanding the discussion on structural stability, binder-free comparisons, and scalability. While the conceptual novelty is incremental, the study provides a practical and well-supported approach toward solvent- and binder-free Li–S cathodes, with convincing experimental validation.

Response: We again sincerely thank the reviewer for the constructive and encouraging feedback. We believe the manuscript has been improved substantially after the revision.

Thank you very much for the valuable comments. Your comments significantly improved the quality of the manuscript.

Reviewer #4

Overall comment: The revisions are substantial, however the following two questions need to be answered:

Response: We sincerely thank the reviewer for the positive and encouraging feedback on our manuscript. In response to the concerns raised, we have carefully revised the manuscript and addressed each point in detail. We hope that these revisions sufficiently clarify the issues and strengthen the manuscript. Point-by-point responses are provided below.

Comment 4-1: Clarify if you are using 5 μm or 5 mm particles in the powder bed simulation, from powder technology viewpoint, because of adhesive force difference, the bulk powder behavior could be totally different.

Response: We sincerely apologize for the typographical error in the manuscript. The particle diameter used in our powder bed punch simulations was indeed 5 mm, not 5 μm which was stated in the Method/Computation modeling for thermal compression behavior of elemental sulfur section. We have corrected this typo in the revised manuscript to ensure consistency and accuracy. We concur with the reviewer's expert insight that particle size is a critical factor influencing the bulk powder behavior, particularly due to the dominance of adhesive forces at the microscale. This is a crucial point in powder technology. However, we wish to clarify the primary objective of our current study. The purpose of our finite element method (FEM) simulations using ANSYS was to gain qualitative insights into the bulk morphological transformation of the powder bed during compaction. Our model was not intended for a direct quantitative comparison with experimental data.

As noted, our simulation employed the Bond model, which considers elastic and viscous forces and moments as a reaction to inter-particle deformations. While the model is capable of capturing a range of contact behaviors, its application at the mesoscale (5 mm) inherently focuses on the macroscopic mechanical response to compression rather than the adhesive-dominant behavior at the true particle scale of the experimental sulfur (<30 μm). At the scale of our simulation, the adhesive forces are negligible compared to the applied compressive forces, so our model effectively captures the bulk behavior governed by contact mechanics and particle rearrangement. The use of a larger, mesoscale particle size in our simulation was a necessary simplification to make the computational model tractable. Simulating the true particle scale with the Bond model would be computationally prohibitive for a model of this size, and it is a common practice to use a representative elementary volume (REV) or a larger, simplified particle size to study macroscopic behavior. To justify the choice of millimeter-scale particle size in our simulation, now we added the following text in the Results and Discussion section.

- In Results and Discussion section of the revised Manuscript:

Note that the simulation is designed for qualitative insights into bulk deformation and densification during powder compaction. To balance accuracy and efficiency, we used millimeter-scale particles, which capture bulk mechanical responses acceptable computational cost, whereas a quantitative comparison with experimental data would require micrometer-scale particles that account for adhesive forces.

- In Methods section of the revised Manuscript:

The simulation domain was sized to include at least ten sulfur particles with a representative diameter of ~5 mm in each spatial direction, ensuring statistically meaningful assessment of bulk powder behavior.

Comment 4-2: Are you using the same bond parameters in the bond model to model the powder compaction behaviour of RT and 80 °C? If this is the case, can you justify what material you need

to describe use bond model in your compaction simulation? is it reasonable to use same baon parameters under same conditions?

Response: We thank the reviewer for the comment. We confirm that we used the same bond parameters in the Bond model for our simulations at both room temperature and 80°C. This choice was deliberate and is justified by the primary purpose of our study, which is to gain qualitative insights into the bulk deformation and densification behavior during powder compaction. In our simulation, the Bond model parameters were used to describe the inter-particle mechanical behavior of the material under compressive loading. We did not use these parameters to describe a specific material; rather, they were selected to capture the general compaction behavior of a ductile, porous powder bed. The focus was on the macroscopic response to applied pressure and the resulting morphological changes. The difference in compaction behavior between RT and 80°C was captured by changing the Young's modulus and Poisson's ratio—the primary material inputs to the simulation—while keeping the bond parameters constant. This approach allowed us to isolate and study the effect of temperature on the material's bulk mechanical response, which is the main objective of our study. To justify the choice of Bond model parameters that are not specific to a certain material, now we added the following text in the Methods section.

- In Methods section of the revised Manuscript:

These Bond model parameters were chosen to represent the general inter-particle mechanics of ductile porous powders, but since they are not material-specific, they cannot be used for direct quantitative comparison with experimental data.

Thank you very much for the valuable comments. Your comments significantly improved the quality of the manuscript.